# Reproducing Arctic springtime tropospheric ozone and mercury depletion events in an outdoor mesocosm sea-ice facility

Zhiyuan Gao[1], Nicolas-Xavier Geilfus[1], Alfonso Saiz-Lopez[2], and Feiyue Wang[1]

[1]Centre for Earth Observation Science, and Department of Environment and Geography, University of Manitoba, Winnipeg, MB R3T 2N2, Canada
[2]Department of Atmospheric Chemistry and Climate, Institute of Physical Chemistry Rocasolano, CSIC, 28006 Madrid, Spain

*Correspondence:* Feiyue Wang (feiyue.wang@umanitoba.ca)

**Abstract.** The episodic build-up of gas-phase reactive bromine species over sea ice and snowpack in the springtime Arctic plays an important role in boundary layer processes, causing annual concurrent depletion of ozone and gaseous elemental mercury (GEM) during polar sunrise. Extensive studies have shown that these phenomena, known as bromine explosion events (BEEs), ozone depletion events (ODEs) and mercury depletion events (MDEs), respectively, are all triggered by gas-phase reactive bromine species that are photochemically activated from bromide via multi-phase reactions under freezing air temperatures. However, major knowledge gaps exist in both fundamental cryo-photochemical processes causing these events and meteorological conditions that may affect their timing and magnitude. Here, we report an outdoor mesocosm study in which we successfully reproduced ODEs and MDEs at the Sea-ice Environmental Research Facility (SERF) in Winnipeg, Canada. By monitoring ozone and GEM concentrations inside large, acrylic tubes over bromide-enriched artificial seawater during sea ice freeze-and-melt cycles, we observed mid-day photochemical ozone and GEM loss in winter in the in-tube boundary layer air immediately above the sea ice surface in a pattern that is characteristic of BEE-induced ODEs and MDEs in the Arctic. The importance of UV radiation and the presence of a condensed phase (experimental sea ice or snow) in causing such reactions was demonstrated by comparing ozone and GEM concentrations between the UV-transmitting and UV-blocking acrylic tubes under different air temperatures. The ability of reproducing BEE-induced photochemical phenomena in a mesocosm in a non-polar region provides a new approach to systematically studying the cryo-photochemical processes and meteorological conditions leading to BEEs, ODEs, and MDEs in the Arctic, their role in biogeochemical cycles across the ocean-sea ice-atmosphere interface, and their sensitivities to climate change.

## 1 Introduction

Every year during springtime in the Arctic, a series of episodic photochemical events is observed concurrently in the boundary layer air, including bromine explosion events (BEEs), ozone depletion events (ODEs), and mercury depletion events (MDEs) (Barrie et al., 1988; Barrie and Platt, 1997; Bottenheim et al., 1986; Oltmans et al., 1989; Oltmans and Komhyr, 1986;

Platt and Hausmann, 1994; Schroeder et al., 1998; Steffen et al., 2005). Subsequent studies have shown that these events are

triggered by the cycling of photolytically activated halogen species (especially bromine species such as Br, BrO, HOBr) over sea ice or snowpack, which rapidly react with ozone and gaseous elemental mercury (GEM) in the boundary layer air (Abbatt et al., 2012; Bognar et al., 2020; Pratt et al., 2013; Saiz-Lopez and von Glasow, 2012; Simpson et al., 2007b; Wang et al., 2019). These annually recurring photochemical processes greatly change the oxidative conditions of the Arctic marine boundary layer during springtime, affecting biogeochemical cycles of many inorganic and organic chemicals across the ocean-

sea ice-atmosphere interface in the Arctic (Wang et al., 2017).

While there is a general consensus on the reaction schemes involved in BEEs, ODEs and MDEs (Fig. 1), major uncertainties exist with respect to the fundamental cryo-photochemical processes causing these events and meteorological conditions that may affect their timing and magnitude. It has been generally assumed that the cycling of reactive bromine species is sustained by HOBr and $BrONO_2$ via multi-phase reactions on the surface of a condensed phase during polar sunrise

(Abbatt et al., 2012; Simpson et al., 2007b, 2015; Wang and Pratt, 2017). Yet the role of HOBr and the nature of the condensed phase remain not well characterized. Some studies have suggested a potential link between bromine activation and the extent of first-year and multi-year sea ice (Bognar et al., 2020; Simpson et al., 2007a), whereas field observations and laboratory studies show that saline snowpack and sea salt aerosols are more likely to provide such a condensed phase for the reactions (Huff and Abbatt, 2002; Pratt et al., 2013; Wang et al., 2019; Wren et al., 2013; Xu et al., 2016). The cycling of bromine

species is favoured under acidic conditions (Halfacre et al., 2019; Pratt et al., 2013), and the surfaces of sea ice and frost flowers, which are highly alkaline (Hare et al., 2013), are unlikely to be effective in sustaining the reactions (Kalnajs and Avallone, 2006). Bromine activation also requires solar radiation, especially in the UV region, and is affected by air temperatures, as BEEs, ODEs and MDEs are only observed during polar sunrise and diminished when the air temperature rises to above 0 °C (Bognar et al., 2020; Burd et al., 2017; Steffen et al., 2005). Atmospheric and sea-state conditions, such as air

mass origin, sea ice and boundary layer dynamics, and blowing snow events, may also affect the timing and magnitude of BEEs, ODEs and MDEs (Bognar et al., 2020; Moore et al., 2014; Thomas et al., 2011; Zhao et al., 2016).

So far most of the studies on BEEs, ODEs and MDEs are based on field observations, which reflect integrated complex processes occurring in nature, but the lack of controllability makes it difficult to address some of the aforementioned knowledge gaps on the fundamental cryo-photochemical processes. The logistics of field campaigns also pose restrictions on

the chemical species that can be measured and the nature and dynamics of the sea ice environment that can be studied. To help elucidate fundamental processes, laboratory studies have been conducted to study the halogen release mechanism from frozen saline solutions (Abbatt et al., 2010; Huff and Abbatt, 2002; Sjostedt and Abbatt, 2008). Such frozen saline solutions may chemically resemble Arctic sea ice or snow substrates, but the laboratory-based studies cannot reproduce the natural growth of sea ice, nor the atmospheric and sea-state conditions. In this study, we present a mesocosm experiment to investigate the

occurrence and magnitude of BEE-induced ODEs and MDEs. The experiment covered several freeze-and-melt cycles of

experimental sea ice and open water periods. The temporal changes in ozone and GEM concentrations were monitored in the boundary layer air mass inside acrylic tubes. Simultaneous photochemical depletions of ozone and GEM under freezing air temperatures were observed in agreement with the characteristics of BEE-induced photochemical phenomena. We show that the experimental design provides a new mesocosm platform to study BEEs, ODEs and MDEs, which will help elucidating

fundamental processes behind their occurrence in the Arctic and their sensitivity to climate change.

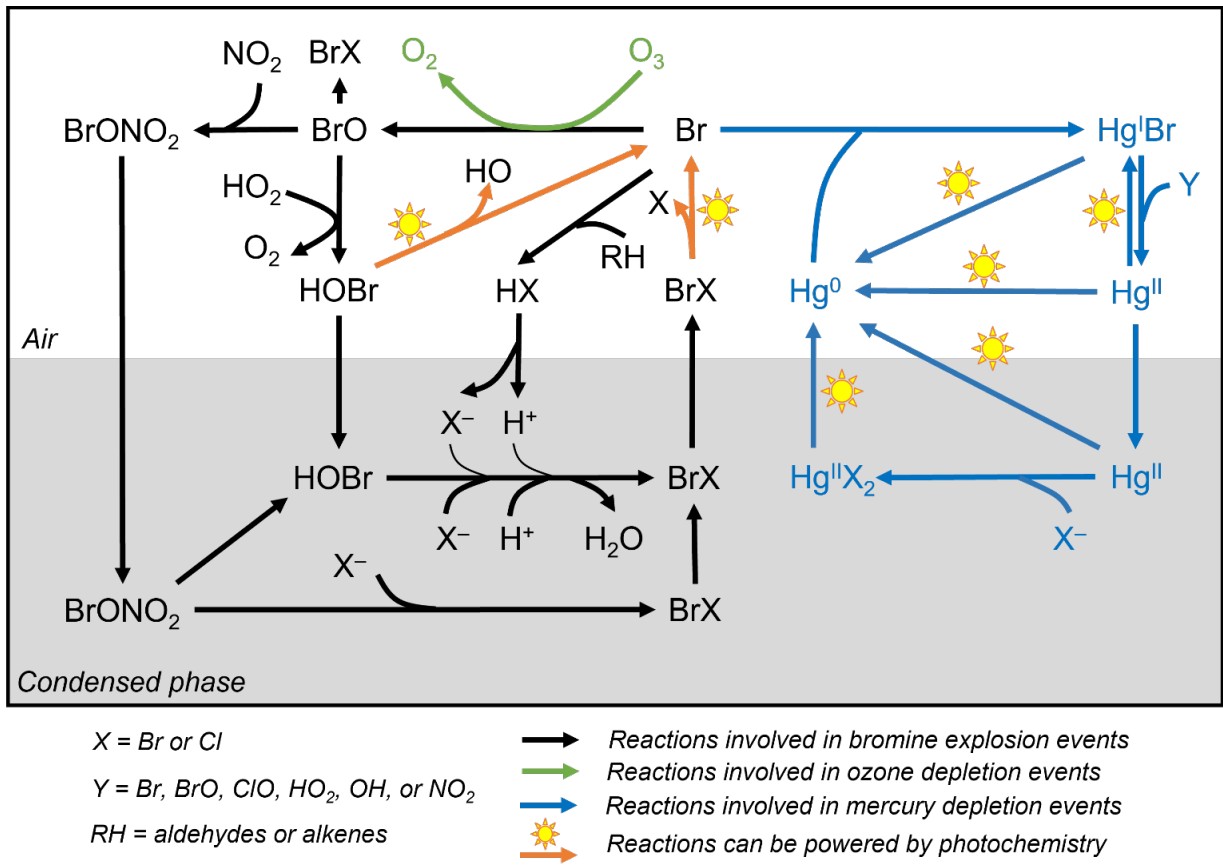

**Figure 1.** General reaction schemes involved in bromine explosion events, ozone depletion events and mercury depletion

events in the Arctic during polar sunrise. The photochemical activation of gas-phase reactive bromine species (Br and BrO) produced from multi-phase reactions on the surface of the condensed phase causes the depletion of ozone and gaseous elemental mercury in the boundary layer air (based on Abbatt et al., 2012; Aguzzi and Rossi, 2002; Khiri et al., 2020; Saiz-Lopez et al., 2018, 2019; Saiz-Lopez and von Glasow, 2012; Simpson et al., 2007b, 2015; Wang et al., 2017, 2019; Wang and Pratt, 2017).

## 2 Methods

### 2.1 Experimental set-up

The mesocosm experiment was carried out in winter (January to March) and fall (October to December) of 2020 at the Sea-ice Environmental Research Facility (SERF; 49.80 °N, 97.14 °W) located on the campus of the University of Manitoba, Winnipeg, Canada, more than 1,800 km south from the Arctic Circle. As the first experimental sea-ice facility in Canada, SERF has been supporting process-oriented mesocosm sea ice studies since 2012. It is equipped with an outdoor pool (18.3 m long, 9.1 m wide and 2.6 m deep) exposed to the ambient environment. The pool is filled with artificial seawater ($\sim$ 380 m$^3$) that resembles the salinity and major ionic composition of Arctic seawater. As the air temperature drops well below 0 °C (down to –30 °C) in winter, the experimental sea ice exhibits similar physical and chemical properties to natural first-year sea ice in the Arctic Ocean (Geilfus et al., 2016; Hare et al., 2013; Xu et al., 2016). The air mass around the SERF facility has typical urban characteristics and may be affected by occasional use of vehicles within the facility.

For this experiment, the artificial seawater was prepared in November 2019 and formulated by mixing groundwater with concentrated NaCl brine and secondary salts ($MgCl_2 \cdot 6H_2O$, $MgSO_4 \cdot 7H_2O$, $CaCl_2$ and NaBr) to achieve a major-ion composition resembling that of the standard seawater (Millero, 2013) at a salinity of 32.8, including chloride (532 mmol kg$^{-1}$), sulphate (28.4 mmol kg$^{-1}$), sodium (495 mmol kg$^{-1}$), calcium (10.3 mmol kg$^{-1}$) and magnesium (33.2 mmol kg$^{-1}$). To create an amplified signal of bromine activation, excess bromide was added to reach a final concentration of 6.5 mmol kg$^{-1}$, which is $\sim$ 8.1 times that (0.8 mmol kg$^{-1}$) of a S = 32.8 seawater. Once prepared, the artificial seawater was left to equilibrate with the atmospheric $CO_2$ for more than one month. To prevent it from freezing, seawater was heated from the bottom of the pool and continuously mixed using circulation pumps until the start of the experiment. At the end of the equilibration (21 January 2020), the artificial seawater had a pH of 7.8, a dissolved inorganic carbon (DIC) concentration of 2500 µmol kg$^{-1}$ and total alkalinity (TA) of 2544 µmol kg$^{-1}$.

In total, four experiments focusing on different ice or open water stages were carried out. Experiments #1 and #2 followed successive freeze-and-melt cycles of experimental sea ice during winter (January to March 2020). Experiment #1 started on 21 January when the heater and circulation pumps were turned off. The ice grew naturally until 22 February when the heating on the bottom of the pool was turned on to melt the ice. Open water first appeared on 24 February at which time the circulation pumps were turned on to speed up the melting process. The pool became completely ice free on 2 March. After the heating and circulation pumps were turned off, Experiment #2 started and continued until 17 March. It followed a natural melting process as the weather warmed up near the end of Experiment #2. Experiment #3 was carried out in October 2020 for two weeks and the pool remained ice free throughout the duration. Experiment #4 was conducted in December 2020 for one week during the early stage of sea ice formation when the air temperature dropped rapidly.

Due to the small surface area of the SERF pool (167 m²) in an otherwise urban environment far away from the Arctic, changes induced by chemical reactions in the boundary layer air over the pool can be greatly interfered by the mixing with the ambient air. To limit such air mixing and be able to monitor the change in ozone concentrations immediately above the seawater or sea ice surface, two large acrylic tubes (inner diameter: 30 cm, height: 183 cm; Emco Industrial Plastics, Cedar Grove, NJ) were fixed into the seawater before it was frozen, with the bottom 50 cm being submerged in the water. They were

placed about 30 cm away from the edge of the pool and were kept vertical by mechanical arms fixed on the edge of the pool (Fig. 2). Both tubes were open to the atmosphere to allow direct air-snow-ice interaction and exchange. One of the tubes is made of UV-blocking acrylic material (cut-off wavelength: 370 nm), and the other of UV-transmitting acrylic material (cut-off wavelength: 270 nm) (see Fig. S1 in the Supplementary Information). On each tube, three brass adaptors were drilled through the acrylic wall as sampling ports on the same side located at 10 cm, 20 cm and 40 cm above the water surface,

permitting real-time gas measurements in the in-tube air at different heights (Fig. 2). For the rest of this paper, the "in-tube air" is referred to the boundary layer air mass above the sea ice surface constrained inside the acrylic tubes, whereas the air outside of the tubes is considered the "ambient air".

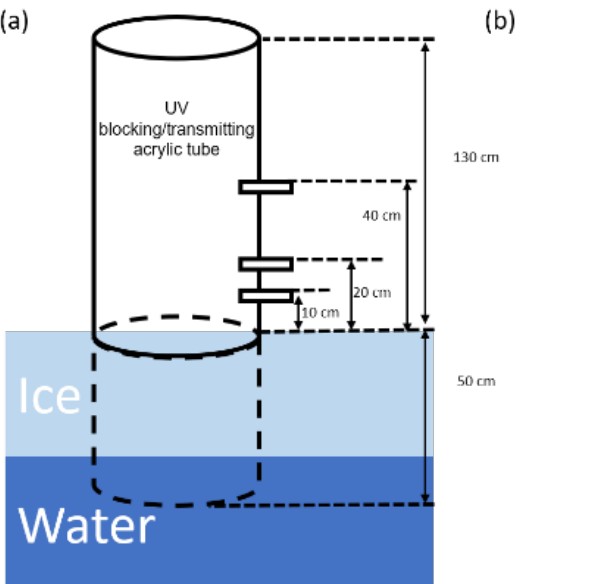 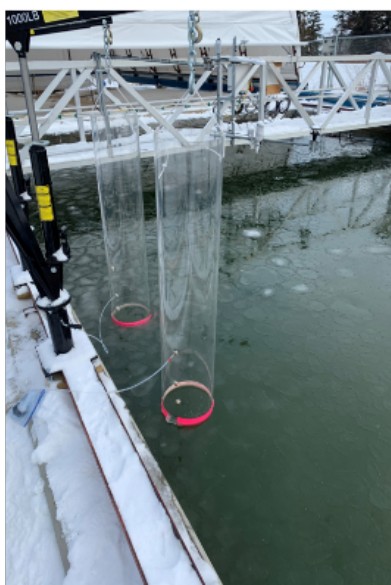

**Figure 2.** (a) An illustration and (b) a photo showing the experimental set-up at the Sea-ice Environmental Research Facility, Winnipeg, Canada. Two acrylic tubes (one UV-blocking and the other UV-transmitting) were fixed vertically in the sea ice pool, with ozone measurement ports drilled at various heights through the tube wall.

## 2.2 Ozone and GEM measurements

Real-time ozone concentrations in the in-tube air and the ambient air were measured using a Teledyne T400 ozone analyzer which reports averaged ozone concentration for every minute with a detection limit of 0.4 ppbv. The flow rate was 0.8 L min$^{-1}$ through a 4-m heated sample line (inner diameter: 4.8 mm). The ozone analyzer was calibrated at the beginning of the experiment, and frequently checked at a span range of 450 ppbv. The relative standard deviation for the span check during the experiment was 2.1 % (n = 11). To allow ozone measurement of the in-tube air between different tubes, a two-way switch (Tekran 1100 dual port module) was used to automatically switch between the sampling ports on different tubes. For example, during 12 to 23 February, continuous sampling was conducted at one height on both UV-transmitting and UV-blocking tubes for more than 40 hr before moving to another height for sampling. During data collection at each height, ozone sampling switched between different tubes at an interval of five minutes. Thus, each ozone data reported herein is averaged over a 5-min integration time (five measurements). The ozone concentrations in the ambient air near the pool were also measured during the experiment.

The quantify and normalize the ozone difference inside the UV-transmitting tube relative to other locations, the ozone loss (%) is reported for every 10 minutes and calculated by Eq (1):

$$\Delta O_3(\%) = \frac{[O_3]_i - [O_3]_{UVT}}{[O_3]_i} \times 100\% \tag{1}$$

where $[O_3]_i$ is the ozone concentration in the ambient air or the in-tube air inside the UV-blocking tube, and $[O_3]_{UVT}$ is the ozone concentration in the in-tube air inside the UV-transmitting tube.

During Experiment #4, GEM was also measured in the in-tube air with a 5-min resolution by cold vapour atomic fluorescence spectroscopy (CVAFS) using a Tekran 2537B mercury analyzer; the detection limit was 0.1 ng m$^{-3}$. The flow rate was 1 L min$^{-1}$ and GEM was monitored using the same sampling system as the ozone measurement. The instrument was routinely calibrated with an internal mercury permeation source and the relative standard deviation was 2.5 % during the experiment (n = 21).

The GEM loss (%) inside the UV-transmitting tube relative to the UV-blocking tube is reported and calculated by Eq (2):

$$\Delta GEM(\%) = \frac{[GEM]_i - [GEM]_{UVT}}{[GEM]_i} \times 100\% \tag{2}$$

where $[GEM]_i$ and $[GEM]_{UVT}$ are the GEM concentrations in the in-tube air inside the UV-blocking tube and inside the UV-transmitting tube, respectively.

## 2.3 Other measurements

In addition to ozone and GEM, nitrogen oxides (NO, NO$_2$ and their sum NO$_x$) concentrations in the ambient air were measured using a Teledyne T200 NO/NO$_2$/NO$_x$ analyzer near the pool during Experiment #2. The instrument reports data for

every minute with a detection limit of 0.4 ppbv. The $NO_x$ data reported in this study is averaged for every 10 minutes, which is the same resolution as the ozone loss ($\Delta O_3$) measurement.

Meteorological conditions were measured at a station located at 1.5 m above the ice surface, including air temperature by a Vaisala HMP45C probe, solar irradiance by a CNR4 net radiometer (Kipp & Zonen, spectral range of 0.3–2.8 µm), and wind by an UltraSonic anemometer (WindSonic). In-situ sea ice and seawater temperatures were measured by an automated type-T thermocouple array installed vertically throughout the depth of the pool at a resolution of 2 cm for the top 50 cm.

Seawater, sea ice and snow samples were collected discretely for salinity, ion composition, pH, DIC and TA analysis. Two ice cores were collected for ion composition analysis using a Mark II coring system (diameter: 9 cm; KOVACS Enterprise, Roseburg, OR) at the middle and end of Experiment #1 on 10 February and 21 February, respectively. The ice core taken on 21 February was also subsampled for pH analysis. More frequent and separate ice core collections were conducted for DIC and TA measurement. All ice cores were taken from outside of the acrylic tubes within a few meters from where the tubes were located and cut into 3-cm sections immediately after the retrieval. Ice sections were stored in gas-tight plastic bags (Nylon/poly, Cabela's) and vacuum-sealed (Hu et al., 2018), followed by melting at 4 °C in the dark until further analysis. Under-ice water was sampled for ion composition analysis by submerging 50-mL polypropylene tubes (Falcon) completely underwater from the hole where the ice core was taken, after manually clearing off floating ice. For DIC and TA measurement, the water samples were collected in 100-mL incubation bottles. Multiple snow samples were taken within 48 hr after the snow deposition by scooping untouched surface snow on sea ice within the pool area and on the nearby land into 50-mL Falcon tubes. Fresh snow on sea ice was collected within 2 hr after deposition for pH analysis. All snow samples were melted in the dark at 4 °C until further analysis.

Seawater, melted sea ice and snow samples were analyzed for ion composition including bromide by ion chromatography (IC) on a DIONEX 5000+ IC system. The recovery and detection limit were determined from repetitive measurements on the least concentrated point of the calibration curve, prepared from a Dionex seven anion standard and a Dionex six cation-II standard respectively, and were 98 % and 1.5 µmol kg$^{-1}$ for bromide, 96 % and 2.0 µmol kg$^{-1}$ for chloride, 99 % and 3.0 µmol kg$^{-1}$ for sulphate, 97 % and 4.6 µmol kg$^{-1}$ for nitrite, 95 % and 1.3 µmol kg$^{-1}$ for nitrate, 107 % and 0.2 µmol kg$^{-1}$ for sodium, 119 % and 0.8 µmol kg$^{-1}$ for magnesium, and 93 % and 1.0 µmol kg$^{-1}$ for calcium. Salinity was calculated from conductivity and temperature (Grasshoff et al., 2007), which was measured by a daily-calibrated conductivity cell probe (Orion 013610MD, Thermo Scientific). The pH measurement was carried out on the meltwater of bulk sea ice and snow samples using a daily-calibrated pH glass electrode (Orion 420A, Thermo Scientific).

The analysis of DIC and TA followed the procedure described in Geilfus et al. (2016). Briefly, ice samples were melted at 4 °C in the dark to minimize the possible dissolution of ikaite crystals. Once sea ice melted, meltwater and seawater were processed similarly by transferring to gas-tight vials (12-mL Exetainers, Labco High Wycombe, UK), preserved with 12 µL saturated $HgCl_2$ solution, and stored at room temperature in the dark until analysis. Total alkalinity was determined by Gran

titration (Gran, 1952) on a TIM 840 titration system (Radiometer Analytical, ATS Scientific) (Hu et al., 2018). The sample (12 mL) was titrated with a standard 0.05 M HCl solution (Alfa Aesar). Dissolved inorganic carbon was measured on a DIC analyzer (Apollo SciTech) by acidifying a 0.75 mL subsample with 1 mL 10 % $H_3PO_4$ (Sigma-Aldrich), followed by quantifying the released $CO_2$ with a $CO_2$ analyzer (LI-COR, LI-7000). Results were then converted from µmol $L^{-1}$ to µmol $kg^{-1}$ based on sample density, which was estimated from salinity and temperature at the time of analysis. Accuracies of $\pm$ 3 and $\pm$ 2 µmol $kg^{-1}$ were determined for TA and DIC, respectively, from analysis of certified reference materials (A.G. Dickson, Scripps Institution of Oceanography, USA).

### 2.4 Statistical analysis

Statistical analysis was carried out using Microsoft Office Excel. One-way ANOVA was used to examine the statistical difference under different occasions and the significant level was set at 0.05 for each test.

## 3. Results

### 3.1 Meteorological, sea ice and snow properties

Solar radiation, wind speed and temperatures of the ambient air, sea ice (when present) and seawater are shown in Figs. 3, 4, and S2. Throughout Experiments #1 and #2, the presence and vertical extent of sea ice is approximated by the –2 °C isotherm (Fig. 3c) in the pool.

During Experiment #1 (24 January to 22 February), the ambient air temperature dropped below –30 °C and the pool surface was completely ice-covered for the entire experiment. The temperature of surface sea ice reached as low as –16 °C on 19 February. Maximal ice thickness (~ 27 cm) was reached on 21 February near the end of Experiment #1. The daytime that is characterized by a positive downward shortwave radiation was ~ 10 hr (from ~ 8:00 to ~ 18:00; all time local = GMT – 6 hr). During Experiment #2 (3 to 17 March), the ambient air temperature increased gradually from –20 °C to –10 °C. The pool surface was only partially ice-covered as a result of warming weather that caused interrupted freezing period. The sea ice (when present) thickness was typically less than 10 cm. A longer daytime was observed (~ 11 hr), from ~ 7:00 to ~18:00. During Experiment #3 (19 to 30 October), no sea ice was observed within the pool. The ambient air temperature ranged from –8 °C to 4 °C, and the daytime lasted ~ 9.5 hr (from ~ 7:30 to ~17:00). During Experiment #4 (9 to 16 December), the pool changed from open water to a complete ice cover and the ice thickness was less than 10 cm. The daytime lasted ~ 7.5 hr (from ~ 8:30 to ~ 16:00). The ambient air temperature dropped rapidly from 18 °C on 9 December to below –10 °C on 14 December.

During Experiments #1 and #2, flurries of snow occurred episodically with less than 1 cm of snow accumulation within the pool area, except on 15 February when up to 5 cm of snow was accumulated on the ice cover outside of acrylic tubes during a snowfall event. The deposited snow went through diel cycle of melting and re-freezing to eventually form a crust

layer on the sea ice surface or was blown away from the SERF pool by wind turbulence within 24 hr after deposition. Yet, the accumulated snow layer (< 2 cm) inside the tubes remained visible for several days on the ice surface and inner tube walls as it was sheltered from the ambient wind. During Experiment #4, up to 4 cm of natural snow accumulation inside the tubes were observed above sea ice after snow flurries occurred on the night of 13 December.

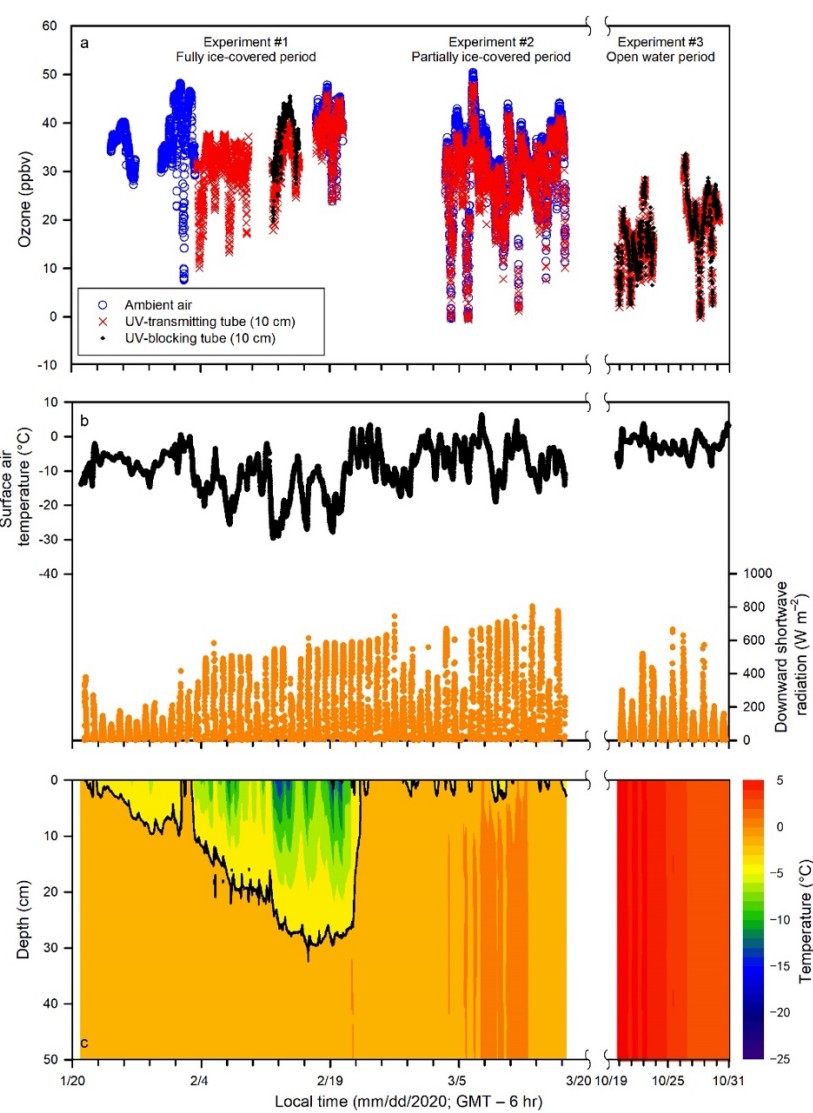

**Figure 3.** Temporal changes of (a) ozone in the ambient air, and the in-tube air inside the UV-transmitting and UV-blocking tubes (10 cm above the ice or water surface); (b) surface air temperature, solar radiation (1.5 m above the ice or water surface), and (c) ice and water temperatures in the top 50 cm of the pool during Experiments #1, 2 and 3.

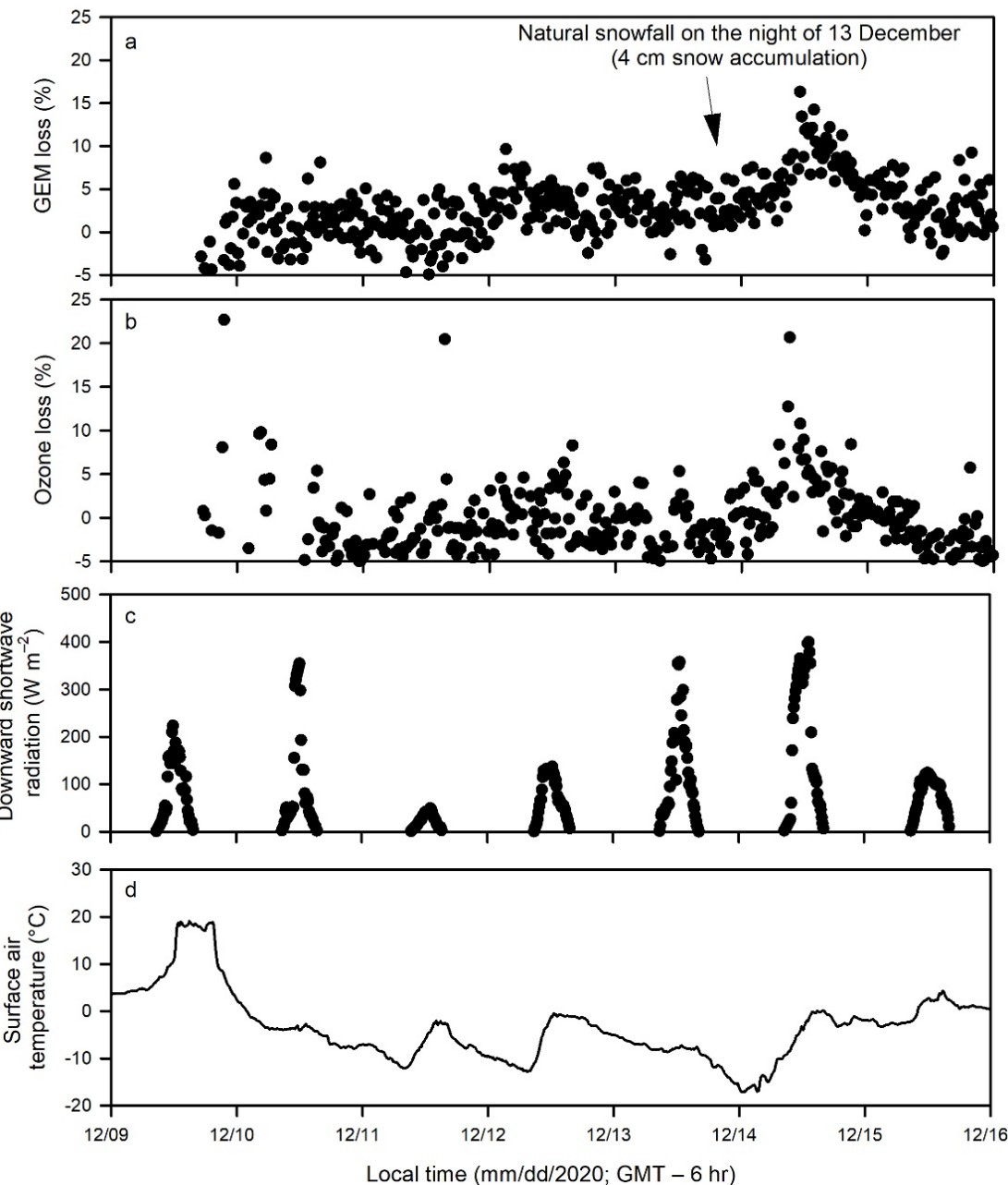

**Figure 4.** Temporal changes of (a) gaseous elemental mercury (GEM) loss and (b) ozone loss (measured as the difference between the UV-transmitting and UV-blocking tubes), (c) downward shortwave radiation, and (d) surface air temperature during Experiment #4.

### 3.2 Ozone in the in-tube air and ambient air

Ozone concentrations in the ambient air and in the in-tube air inside UV-transmitting and UV-blocking tubes (at 10 cm above the ice surface) are shown in Fig. 3. Large temporal variations in ozone concentrations were found in both the in-tube air inside the UV-transmitting tube and ambient air, ranging from 0 to 50 ppbv, yet they showed similar temporal patterns. During the overlapped data collection period (17 to 20 February and 3 to 17 March), ozone in the ambient air ($33 \pm 9$ ppbv; mean $\pm$ standard deviation) was slightly but significantly higher than that in the in-tube air measured at 10 cm above the ice surface inside the UV-transmitting tube ($31 \pm 8$ ppbv) ($p < 0.05$). The averaged ozone difference between the UV-transmitting tube and ambient air was $2 \pm 2$ ppbv for the entire experiment when measurements were done.

Comparisons of ozone concentrations in the in-tube air between UV-transmitting and UV-blocking tubes are shown in Fig. 5. Ozone was measured at different heights (10, 20, 40 cm) above the sea ice surface during Experiment #1, whereas during Experiment #3 all measurements were done at 10 cm above the water surface. The ozone concentrations in the in-tube air between the two tubes were not significantly different when measured at 40 cm ($p > 0.05$) and 20 cm ($p > 0.05$) above the sea ice surface during Experiment #1 (Fig. 5a, b), or at 10 cm above the water surface during Experiment #3 ($p > 0.05$) (Fig. 5c). The only exception is the measurement conducted closest to the sea ice surface (10 cm) during the ice-covered period in Experiment #1, when the ozone concentration in the UV-transmitting tube was significantly lower than that in the UV-blocking tube ($p < 0.05$) (Fig. 5c). The averaged ozone difference between two acrylic tubes at 10 cm above the sea ice surface was $5 \pm 2$ ppbv.

### 3.3 Simultaneous depletion of GEM and ozone in the in-tube air

Both ozone and GEM were measured during Experiment #4 (December 2020). The depletions of GEM (Fig. 4a) and ozone (Fig. 4b) occurred in the same pattern and at the same time. In addition, a larger extent of depletion of both GEM and ozone was observed on 14 December following the accumulation of a thin snow layer ($\sim$ 4 cm) above sea ice inside both acrylic tubes from the precipitation in the previous night.

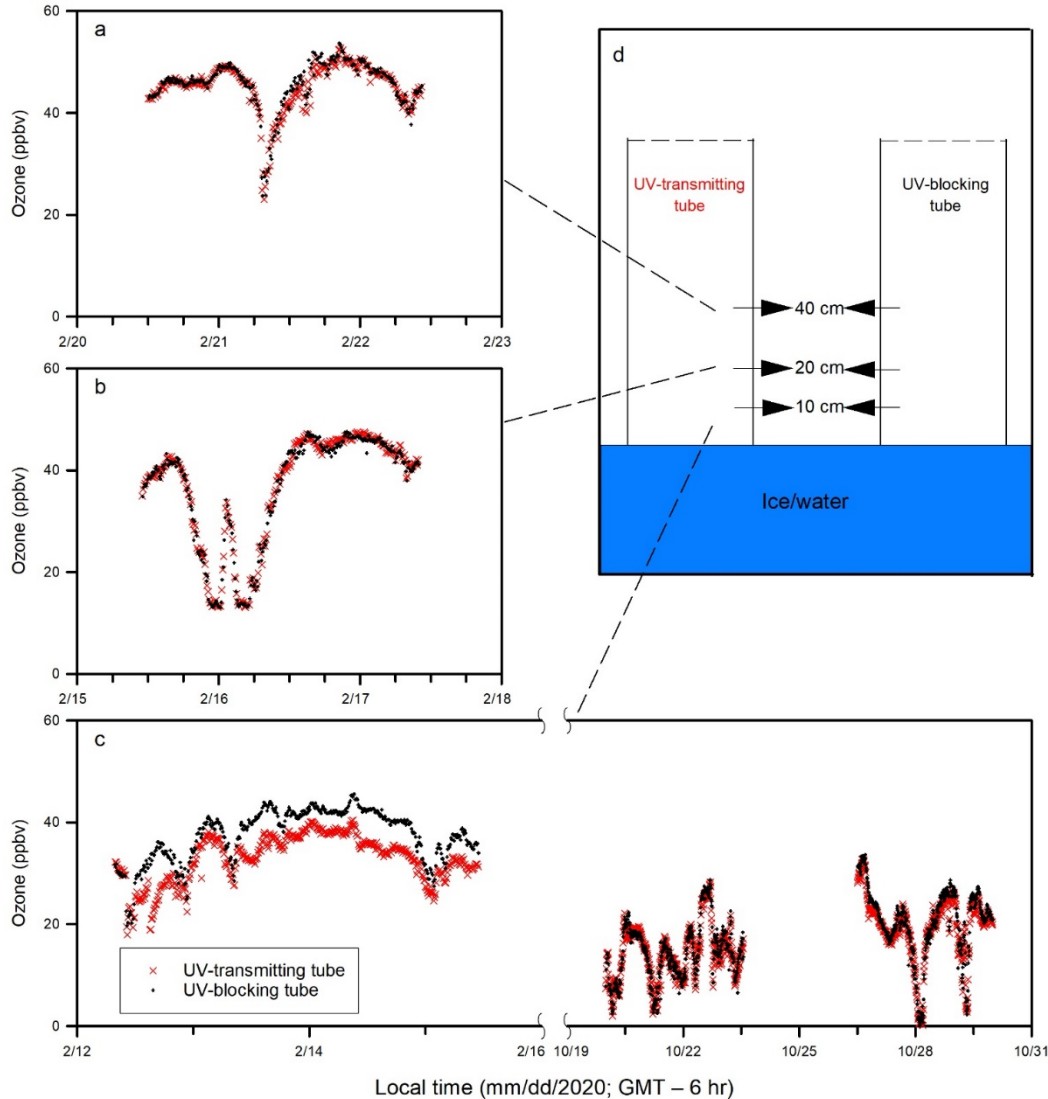

**Figure 5.** Ozone concentrations in the in-tube air inside the UV-transmitting and UV-blocking tubes measured at (a) 40 cm, (b) 20 cm, and (c) 10 cm above the ice (or water) surface. The measurement schemes are illustrated in (d).

## 3.4 Chemical composition of sea ice and snow

The mesocosm experiment was conducted using bromide-enriched artificial seawater. As expected, considerable amounts of salts from seawater are retained in sea ice (Table 1). Ion concentrations are very low in snow collected from nearby land

surfaces, whereas considerably (~ 100 times) higher concentrations are found for the thin layer of snow above sea ice, which is consistent with the brine-wetting process in snow overlying sea ice (Barber and Nghiem, 1999). The exceptions are nitrite and nitrate. Both ions are found to be below the detection limit for almost all the samples except the snow collected from nearby land surfaces. The measured concentrations of bromide and chloride in saline snow and surface sea ice samples were much higher than the values previously reported for Arctic snow samples over first-year and multi-year sea ice (Krnavek et al., 2012; Peterson et al., 2019). This large difference can be explained by a dominant contribution from sea ice brine in our mesocosm experiment, and a more prevalent atmospheric source of halides in natural Arctic snowpack (Peterson et al., 2019). Bromide was found to be preferentially enriched in sea ice and in the overlying snow, as demonstrated by elevated $Br^-/Cl^-$ mole ratio (0.02) when compared with that in the underlying seawater (0.01). Similar preferential enrichment was not observed for other major ions.

The vertical profile of pH across the snow-sea ice-seawater interface was measured on the ice core taken on 21 February (Fig. 6). The pH of the bulk sea ice meltwater changed from 8.1 at the bottom, to 8.0 in the mid-section, and as high as 9.4 near the surface. Estimate based on the measurements of salinity, temperature, DIC and TA (Table 2) suggests that the thin ice and the surface layer of the growing ice consistently had a pH > 8.5 during Experiment #1. Both the high pH values near the surface and the C-shaped vertical distribution pattern are in good agreement with those reported in a previous mesocosm study at SERF (Hare et al., 2013). The snow overlying sea ice had a pH of 7.2. It is not as acidic as expected from that of fresh snow (6.7) due to the influence of sea ice brine, but still more than one to two orders of magnitude more acidic than the surface sea ice.

**Table 1.** Ion composition of snow, surface ice and surface seawater during Experiment #1. DL: Detection limit calculated as 2.998 times the standard deviation determined from eight repetitive measurements of the least concentrated point of the calibration curve.

| | Concentration (mmol kg$^{-1}$) | | | | | | | | Mole ratio | | | | | |
|---|---|---|---|---|---|---|---|---|---|---|---|---|---|---|
| | $Cl^-$ | $Br^-$ | $SO_4^{2-}$ | $NO_2^-$ | $NO_3^-$ | $Na^+$ | $Mg^{2+}$ | $Ca^{2+}$ | $Br^-/Cl^-$ | $Cl^-/Na^+$ | $Br^-/Na^+$ | $SO_4^{2-}/Na^+$ | $Mg^{2+}/Na^+$ | $Ca^{2+}/Na^+$ |
| Snow over land (n = 8) | 1.5 ± 0.8 | 0.002 ± 0.001 | 0.01 ± 0.00 | < DL | 0.019 ± 0.005 | 0.4 ± 0.2 | 0.02 ± 0.02 | 0.06 ± 0.02 | 0.001 ± 0.000 | 4.8 ± 2.6 | 0.004 ± 0.001 | 0.04 ± 0.00 | 0.06 ± 0.03 | 0.3 ± 0.1 |
| Snow over sea ice (n = 11) | 290 ± 105 | 4.3 ± 2.1 | 16.5 ± 5.6 | < DL | < DL | 261 ± 96 | 19.8 ± 9.8 | 5.8 ± 2.2 | 0.02 ± 0.01 | 1.1 ± 0.0 | 0.02 ± 0.01 | 0.07 ± 0.02 | 0.07 ± 0.03 | 0.02 ± 0.00 |
| Sea ice (top 3 cm) (n = 4) | 260 ± 23 | 4.9 ± 1.7 | 17.9 ± 0.8 | < DL | < DL | 251 ± 20 | 18.1 ± 3.0 | 5.0 ± 0.6 | 0.02 ± 0.01 | 1.1 ± 0.0 | 0.02 ± 0.01 | 0.07 ± 0.00 | 0.07 ± 0.01 | 0.02 ± 0.00 |
| Surface seawater (n = 3) | 532 ± 8 | 6.5 ± 1.1 | 28.4 ± 1.1 | < DL | < DL | 495 ± 2 | 33.2 ± 16.6 | 10.3 ± 0.2 | 0.01 ± 0.00 | 1.1 ± 0.0 | 0.01 ± 0.00 | 0.06 ± 0.00 | 0.07 ± 0.03 | 0.02 ± 0.00 |

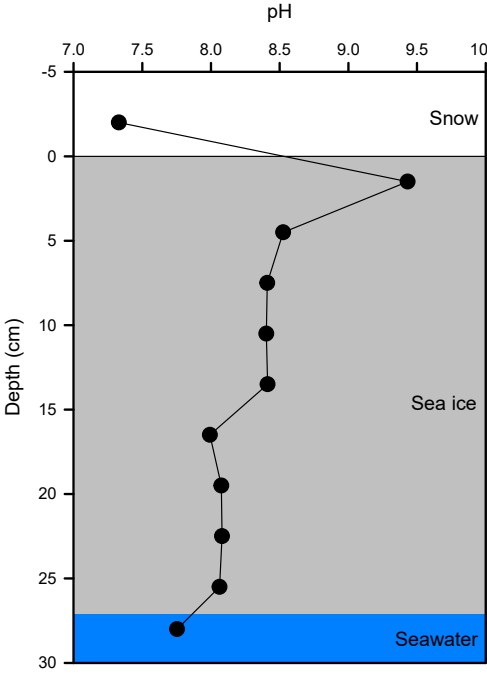

**Figure 6.** Vertical profile of pH across the snow-sea ice-seawater interface, as measured on the ice core taken on 21 February.

**Table 2.** Dissolved inorganic carbon (DIC) and total alkalinity (TA) of the seawater and sea ice during Experiment #1

| Date | Depth (cm) | Salinity | TA (µmol kg⁻¹) | DIC (µmol kg⁻¹) |
|------|-----------|----------|----------------|-----------------|
| *Seawater:* | | | | |
| 21 January | | 32.8 | 2544 | 2500 |
| *Sea ice:* | | | | |
| 22 January | bulk | 15.6 | 1256 | 1267 |
| 24 January | bulk | 13.4 | 1053 | 877 |
| 27 January | bulk | 12.3 | 994 | 438 |
| 30 January | 0 – 4 | 5.6 | 484 | 824 |
| 5 February | 0 – 5 | 7.3 | 609 | 518 |
| 7 February | 0 – 5 | 8.0 | 692 | 623 |
| 10 February | 0 – 5 | 7.4 | 670 | 532 |
| 14 February | 0 – 5 | 9.2 | 759 | 604 |
| 20 February | 0 – 5 | 6.3 | 519 | 491 |
| 24 February | 0 – 5 | 10.3 | 844 | 790 |

## 4. Discussion

### 4.1 Cryo-photochemically driven ozone loss in the in-tube air

The temporal variations of ozone concentrations in the ambient air at SERF (Figs. 3a and 7a) agree well with those
reported for Canadian cities (Angle and Sandhu, 1989; Raddatz and Cummine, 2001). On a diurnal basis (Fig. 7a), the ambient ozone concentrations increase gradually after sunrise (~ 7:30), and peak around 15:00 to 17:00, corresponding to the production of ozone during photochemical oxidation of hydrocarbons from automobile exhaust, which is also supported by the variation in downward shortwave radiation (Fig. 7c). After sunset, the ambient ozone concentrations increase slightly due to the inflow of ozone from surrounding rural areas as a result of nocturnal urban heat island effect (Raddatz and Cummine, 2001). In
addition, the presence of $NO_x$ would also affect ambient ozone dynamics (Fig. S3) as occasional ozone troughs (< 10 ppbv) during night in the ambient air were observed with NO peaks (Fig. S3a). Similar diurnal patterns of the ozone concentrations are evident in the ambient air and the in-tube air inside the UV-transmitting tube even at 10 cm above the sea ice surface (Fig. 3a and Fig. 7a), suggesting that the ozone concentration in the in-tube air is largely controlled by the urban signal (i.e., the ambient air). However, ozone concentrations in the ambient air are considerably higher than that inside the UV-transmitting
tube during sun-lit, daytime (Fig. 7a). This could be indicative of limited mixing of the ambient air inside the tube due to the wall effect, influence on ozone dynamics from $NO_x$ chemistry, and/or loss of ozone inside the tube due to the presence of the experimental sea ice.

To address which of these processes is primarily responsible for the observed ozone difference, we investigate the influence of $NO_x$ chemistry and compare the ozone concentrations measured inside the UV-blocking and UV-transmitting
tubes. The temporal pattern of the ozone difference between the in-tube air inside the UV-transmitting tube and the ambient air (Fig. S3) shows the influence of $NO_x$ on the ozone loss ($\Delta O_3$) within the in-tube air is limited. For example, from 11 to 14 March, NO peaks (20 ppbv) were observed on 11 March and 14 March, whereas NO stayed at relative low levels (< 3 ppbv) during 12 to 13 March. Yet, during the same time, $\Delta O_3$ within the in-tube air reoccurred daily in a diurnal pattern and reached a similar extent regardless of NO concentrations. Additional discussion and figures on the potential influence of $NO_x$ chemistry
on $\Delta O_3$ within the in-tube air can be found in the Supplementary Information. On the other hand, the in-situ $NO_x$ production via snowpack photochemistry of nitrate and nitrite is considered negligible due to the low amount of both ions (below the detection limit) found in surface sea ice and saline snow samples. Thus, $NO_x$ produced either from urban transportation (in the background ambient air) or snowpack photochemistry had negligible influence on $\Delta O_3$ within the in-tube air (i.e., ozone difference between the ambient air and in-tube air inside the UV-transmitting tube).

The potential ozone difference caused by the presence of experimental sea ice is further examined by the comparison of ozone concentrations between UV-blocking and UV-transmitting tubes. The overall temporal patterns between the two tubes were similar during Experiment #1 (Fig. 5). Generally, the ozone concentrations in the in-tube air immediately (10 cm) above

the sea ice inside the UV-transmitting tube were considerably lower than those in the UV-blocking tube (Fig. 5c). The associated ozone loss ($\Delta O_3$) showed a clear diurnal pattern with the largest difference (> 25%) appearing in the early afternoon
(12:00 to 15:00) that corresponded to the peak time of the downward shortwave radiation (Fig. 8). Since the major difference between the two tubes is their UV-transmitting ability in the range of 270 to 370 nm (Fig. S1) and both tubes were open to the same ambient air mass, the considerable mid-day ozone loss inside the UV-transmitting tube can only be attributed to photochemical processes that are prohibited inside the UV-blocking tube.

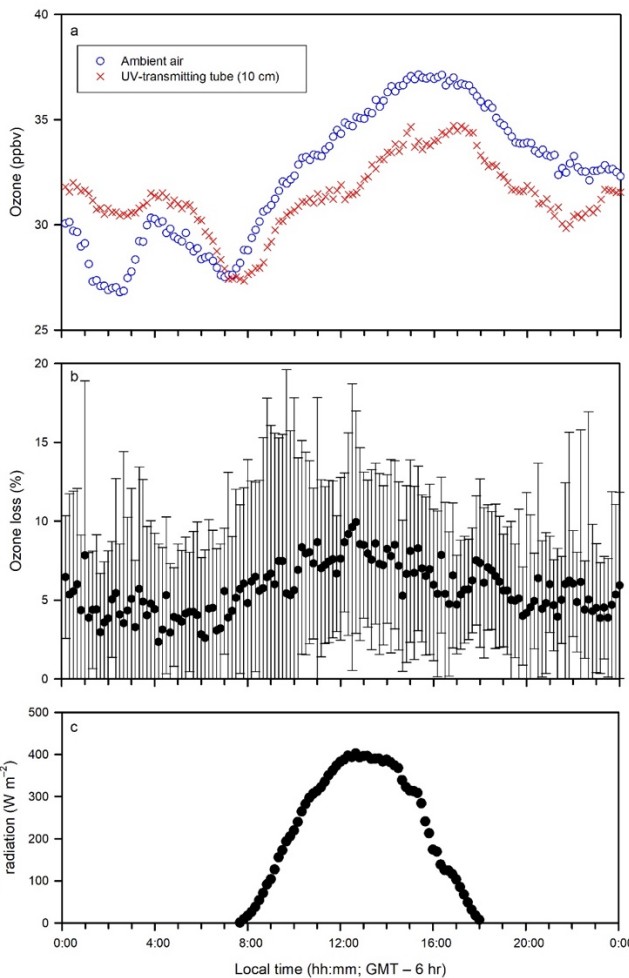

**Figure 7.** Diurnal patterns (averaged over 24 January to 17 March whenever the measurement was available) of (a) ozone concentrations in the ambient air and the in-tube air inside the UV-transmitting tube (10 cm above the sea ice), (b) ozone loss (%) (measured as the ozone difference between the ambient air and the in-tube air inside the UV-transmitting tube), and (c) downward shortwave radiation.

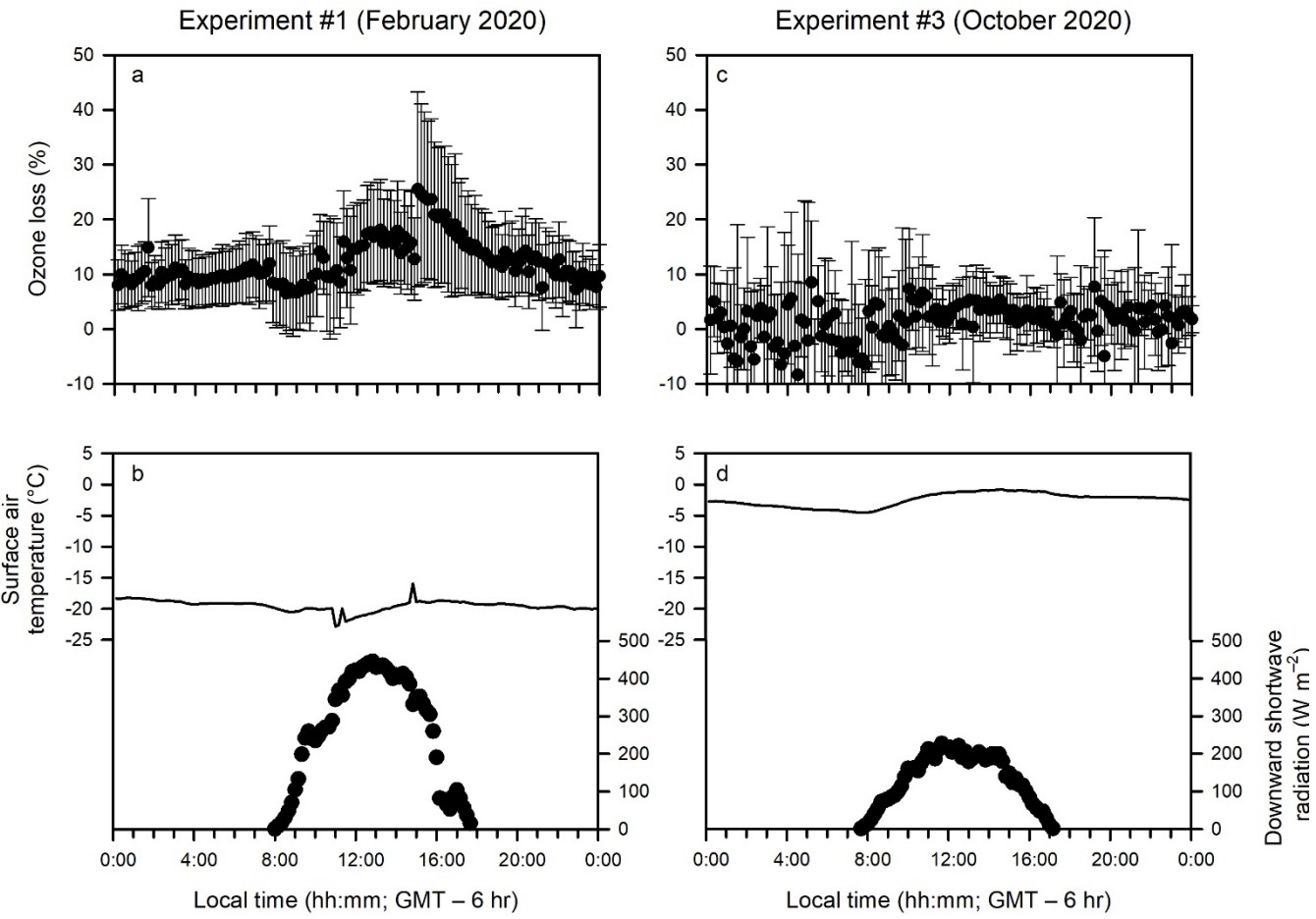

**Figure 8.** Diurnal patterns of (a, c) ozone loss (%) (measured as the ozone difference in the in-tube air between the UV-transmitting and UV-blocking tubes at 10 cm above the sea ice or water surface), and (b, d) meteorological conditions during the ice-covered (a, b) and open water (c, d) experiments.

The observation that no such ozone loss occurred in the in-tube air when measured farther away (20 cm and 40 cm) from the sea ice surface (Fig. 5a, b) suggests that the ozone loss between two tubes is not associated with $NO_x$ photochemistry or ozone photolysis, which should occur universally throughout the tube. Instead, it is most likely triggered by cryo-photochemical processes that involve the sea ice environment. At 20 cm or 40 cm above the sea ice surface, the ozone difference between the two tubes becomes indiscernible due to either the faster mixing of the ozone-rich ambient air or the

lack of ozone-depleting processes away from the sea ice surface. The importance of cryo-photochemical processes in the ozone loss is further supported by Experiment #3 when the measurement was conducted in the absence of sea ice, and no ozone loss was observed even immediately (10 cm) above the water surface (Fig. 5c and Fig. 8c).

Since the UV-transmitting tube only has a UV transmittance of ~ 60% (Fig. S1), even higher magnitude of the ozone loss is expected to occur outside of the acrylic tubes over the entire SERF sea ice surface. However, the rapid mixing of the boundary layer air and the ambient air would readily bury the ozone-depleting signal under the much larger background of ambient urban air, making it difficult to directly observe such cryo-photochemical ozone loss. The use of the acrylic tubes is thus critical for the observation, as they limited the air mixing and permitted gas sampling from the in-tube air that was directly affected by the sea ice surface. Moreover, comparing ozone concentrations between tubes cancels out the variations that exist in the ambient air (e.g., background $NO_x$ produced from urban transportation) since both tubes were open to the same air mass and the ozone dynamics in the ambient air should be equally applied to both tubes.

It should also be noted that the relationship between the intensity of downward shortwave radiation and the extent of the cryo-photochemical ozone loss is yet to be validated (Fig. 4 and Fig. S6). The lack of correlation may be explained by the fact that the downward shortwave radiation reported in this study covers wavelength from 300 nm to 2800 nm, whereas UV radiation (< 400 nm) has been generally considered the most effective range causing photochemical ozone loss. Other factors, such as the presence of various condensed phase, the availability of $Br^-$ substrates and air temperatures would also affect the extent of the cryo-photochemical ozone loss.

## 4.2 Mesocosm reproduction of Arctic springtime photochemical phenomena

Although no direct measurements were made in this study on gas-phase bromine species in the air above sea ice or snow, we believe the simultaneous cryo-photochemical ozone and GEM loss observed over the SERF pool (Fig. 4) are caused by the same mechanism involving bromine activation that is responsible for the Arctic springtime ODEs and MDEs. The other known tropospheric ozone depletion mechanisms include direct photolysis (Wu et al., 2007), photochemical $NO_x$ reactions (Nakayama et al., 2015), and ozone deposition on the tube walls, which are negligible at the study site as demonstrated by a lack of ozone loss during the early fall (Fig. 8c).

As shown in Fig. 1, Br radicals-induced cryo-photochemical ozone and GEM loss involve multi-phase reactions on the surface of a condensed phase under acidic conditions. The condensed phase can be either the bare sea ice or the thin layer of snow accumulated on the sea ice surface at the second half of Experiment #1 and during Experiment #4. The measured $Br^-$/$Cl^-$ ratios in saline snow and in surface sea ice samples (0.02) are found to be in favour of active cycling of bromine species. Pratt et al. (2013) and Peterson et al. (2019) suggested $Br_2$ production from snow is enhanced above a $Br^-$/$Cl^-$ mole ratio of 0.005, which is below our observed ratios for the potential condensed phase reactors. However, the highly alkaline nature of the sea ice surface (Fig. 6; see also Hare et al., 2013, Kalnajs and Avallone, 2006) suggests it is not the most efficient surface

where bromine activation takes place. Instead, the overlying snow is more likely acting as a more favourable condensed phase for the reactions to take place. The importance of the snow layer was further demonstrated during Experiment #4 when stronger ODEs and MDEs were observed shortly after the accumulation of the snow layer on sea ice. The snow layer overlying sea ice is more acidic (Fig. 6) yet enriched in bromide (Table 1) due to the upward migration of sea ice brine. As shown in Table 1, the bromide added in the artificial seawater eventually contributed to enriched bromide concentrations found in sea ice surface and the overlying snow layer, providing abundant substrates for bromine activation. Furthermore, the occurrence of BEEs is indicated by the observation of concurrent ODEs and MDEs (Fig. 4a, b) in the mesocosm.

## 5. Conclusion

In this paper, we show that the Arctic springtime ODEs and MDEs can be reproduced in an outdoor, mesocosm sea ice facility located in an urban area far away from the Arctic. By constraining the boundary layer air mass within acrylic tubes, cryo-photochemical ozone and GEM loss above experimental sea ice are observed with a diurnal pattern that is characteristic of BEE-induced photochemical phenomena. Meanwhile, simultaneous occurrences of both ODEs and MDEs are observed in the in-tube air near the sea ice and snow surface. The comparison between the UV-blocking and UV-transmitting acrylic tubes further emphasizes the role of UV radiation (270 to 370 nm) in causing these photochemical phenomena.

Several improvements can be made for future studies. Even though organic matter and biota were not intentionally introduced into the mesocosm, some of them could have made into the system; to which extent they could have influenced the cryo-photochemical processes (e.g., via the production of bromocarbons) warrants further exploration. Another aspect is related to ice-nucleating particles that are likely to affect the roughness of the sea ice surface and assist in frost formation on the tube walls during mornings, which could potentially act as a temporary condensed phase for bromine activation just after sunrise. Time-series measurements of pH at the air-ice/snow interface could also be explored to probe the availability of protons and the efficiency of HOBr dissociation on the surface of sea ice or snow layer.

The success in reproducing BEE-induced photochemical phenomena in a mesocosm experiment provides a new approach that can supplement, bridge and integrate the laboratory and field-scale studies to advance our understanding of the cryo-photochemical processes and meteorological conditions leading to BEEs, ODEs, and MDEs in the Arctic. In addition to much simpler logistical preparations, major advantages of the mesocosm approach are the ability to control and modify the nature and dynamics of various condensed phases (e.g., sea ice, frost flowers, open leads, overlying snow and blowing snow under various growth and melting conditions), the ability to dope or alter the chemical composition of seawater and condensed phases (e.g., addition of bromide and mercury stable isotopes, acidity), the ability to control the light environment, and the accessibility to various monitoring and analytical capacities (e.g., those for bromocarbons, ice-nucleating particles) that are difficult to access in the remote Arctic. The results will not only help fill up critical knowledge gaps related to BEEs, ODEs and MDEs,

but also aid the development and parameterization of mechanistic models to allow better projection of their sensitivities to climate change in the Arctic, and better understanding of implications for biogeochemical cycles across the ocean-sea ice-atmosphere interface.

## Data availability

The datasets generated in this study are available on request from the corresponding author.

## Author contribution

ZG and FW designed the experiments and led the data interpretation. ZG and NXG carried out the experiments. ASL assisted in data interpretation. ZG and FW prepared the manuscript with contributions from all co-authors.

## Competing interests

The authors declare that they have no conflict of interest.

## Acknowledgements

This work was funded by the Natural Science and Engineering Council of Canada (NSERC), the Canada Research Chairs program, the Canada Excellence Research Chairs (CERC) program, and the University of Manitoba Graduate Fellowship. We thank D. Armstrong for assistance in the ion chromatography analysis. We thank D. Binne and R. Galley for assistance in the experimentation at the Sea-ice Environmental Research Facility (SERF). We thank two anonymous referees and Editor J. Kuttippurath for their constructive comments during the peer-review process.

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
