# Peer review of "Reproducing Arctic springtime tropospheric ozone and mercury depletion events in an outdoor mesocosm sea-ice facility"

_Atmospheric Chemistry and Physics, 2021_

## Author Comment (AC1)

Dear Referee #1,

Thank you very much for reviewing our manuscript and for your constructive and valuable comments and suggestions. In the following text, our response to your comments is given in blue, whereas the corresponding revisions in the main text are highlighted in red.

Gao et al present a study of ozone loss near the surface of an outdoor mesocosm sea ice facility in Winnipeg, Canada. The sea ice facility is unique and provides an opportunity to control the formation of the sea ice with complete exposure to the atmosphere, allowing ambient snow accumulation. The comparison of O3 levels during daytime for the UV-transmitting vs UV-blocking tubes is useful. The authors
attribute O3 loss at 10 cm above the sea ice surface (observed for the UV-transmitting but not UV-blocking, showing that this is a photochemical process) to reaction with bromine radicals based on their enhancement of seawater Br-, which then migrated into the snow above. My major comment is the lack of discussion of the role of NOx, which is known to be important in O3 and bromine chemistry, and
should be particularly important in this urban ambient study (see detailed comments below). This is particularly important because snow photochemical reactions produce NO, which can react with O3, and so disentangling this from reaction with Br is important to consider. Additionally, there is additional published literature, stated below, that is relevant and should be considered in this
manuscript.

The role of NOx needs to be discussed in this study, as there is currently no mention of NOx in the paper. NO reacts with O3, and it seems possible that this could be contributing in part to the O3 loss observed over snow (see, for example, Peterson and Honrath 2001, Geophys. Res. Lett., "Observations of rapid photochemical
destruction of ozone in snowpack interstitial air"), as NOx is released from the snowpack from snow nitrite and nitrate photolysis. Peterson and Honrath (2001) calculated the fraction of the observed ozone loss rate that could be attributed to NOx (they found it was small for their study and then considered bromine reaction), and this seems important to consider here. Were snow nitrate and nitrite measured?
If so, this seems important to report. Further, in the urban environment of this study, NOx is likely elevated due to combustion emissions, especially since the authors discuss O3 formation (involving hydrocarbons and NOx) from vehicle exhaust.

Response: We agree that the role of $NO_x$ should be discussed. Ambient concentrations of $NO_x$ (both NO
and $NO_2$) were indeed measured during Experiment #2 from the same gas sample line as the ambient $O_3$ measurement. The concentrations of $NO_x$, $O_3$ as well as $O_3$ loss (%) within the in-tube air (difference between ambient air and UV-transmitting in-tube air) are provided in Figure 1 below. The range for $NO_x$ (NO and $NO_2$) was capped at 100 ppb in Figure 1a, although some extremely high values (up to 300 ppb) were observed due to vehicle activities that were close to the pool. With respect to $NO_x$ influence on
ambient $O_3$, the concurrence of NO peaks (Fig 1a) was observed with $O_3$ troughs in ambient air (Fig. 1b), especially when ambient $O_3$ dropped below 10 ppb. On the other hand, $NO_x$ influence on $O_3$ loss (%) within the in-tube air is less consistent. For example, from 11-14 March, NO peaks (20 ppb) were observed on 11 March and 14 March, whereas NO stayed at relative low levels (< 3 ppb) during 12-13 March. Yet, during the same time (11-14 March), $O_3$ loss (%) within the in-tube air reoccurred daily in
a diurnal pattern and reached a similar extent regardless of NO concentrations. In the original manuscript, this $O_3$ loss (%) within the in-tube air was attributed to the retardant air mixing rates due to the tube effect or enhanced signals from chemical reactions occurred near ice surface. The observed patterns during 11-14 March suggest a more dominant role of photochemical reactions (e.g., Br chemistry) near ice surface and a lesser influence from ambient NO concentrations.

Both nitrate and nitrite were measured in snow samples by ion chromatography as part of the major ion analysis. Nitrite was always below method of detection limit (MDL) of 4.6 μmol kg$^{-1}$. Nitrate concentrations in most snow samples were also below MDL of 1.3 μmol kg$^{-1}$, except for the land snow samples, which were $19 \pm 5$ μmol kg$^{-1}$. Nitrate in land snow is likely sourced from atmospheric deposition, which is originally produced from vehicle exhaust, because nitrate was not intentionally added in our
artificial seawater. Due to the very low concentrations of nitrate and nitrite found in the snow above sea ice, it is likely that in-situ NO$_x$ production via snowpack photochemistry is limited within the in-tube air mass.

Furthermore, the photochemical O$_3$ loss we present in the manuscript is obtained by comparing O$_3$ concentrations between the two different acrylic tubes. This comparison helps to cancel out the O$_3$ variations in ambient background (caused either by urban signal or NO$_x$ chemistry) since both tubes were open to the same air mass. Thus, we think NO$_x$, either from ambient air or snowpack photochemistry, has limited influence on the O$_3$ depletion patterns discussed in this manuscript. In the revised manuscript, we will add NO$_x$ discussion on O$_3$ dynamics and Figure 1 will be included in the supplement materials.

**Figure 1.** Temporal changes of (a) NO$_x$ (NO and NO$_2$) in ambient air; (b) ozone in ambient air, and (c) ozone loss (measured as the difference between in-tube air and ambient air) during Experiment #2. (This figure will be added as Figure S3 in the revised manuscript)

Some of the revisions in the main text are provided here: the following changes will be made in Lines 145-150 (Section 2.3): "The recovery and method of detection limit (MDL) were determined from repetitive measurements on a Dionex seven anion standard and a Dionex six cation-II standard, and were 98 % and 1.46 µmol kg$^{-1}$ for bromide, 96 % and 1.96 µmol kg$^{-1}$ for chloride, 99 % and 3.05 µmol kg$^{-1}$ for sulphate, 97 % and 4.59 µmol kg$^{-1}$ for nitrite, 95 % and 1.27 µmol kg$^{-1}$ for nitrate, 107 % and 0.20 µmol kg$^{-1}$ for sodium, 119 % and 0.79 µmol kg$^{-1}$ for magnesium, and 93 % and 0.95 µmol kg$^{-1}$ for calcium."

Table 1 will also be revised:

**Table 1.** Major-ion composition of snow, surface ice and surface seawater during Experiment #1

| | Concentration (mmol kg$^{-1}$) | | | | | | | | Molar ratio | | | | | |
| --- | --- | --- | --- | --- | --- | --- | --- | --- | --- | --- | --- | --- | --- | --- |
| | $Cl^-$ | $Br^-$ | $SO_4^{2-}$ | $NO_2^-$ | $NO_3^-$ | $Na^+$ | $Mg^{2+}$ | $Ca^{2+}$ | $Br^-/Cl^-$ | $Cl^-/Na^+$ | $Br^-/Na^+$ | $SO_4^{2-}/Na^+$ | $Mg^{2+}/Na^+$ | $Ca^{2+}/Na^+$ |
| Snow over land (n=8) | 1.5 ± 0.8 | 0.002 ± 0.001 | 0.01 ± 0.00 | < MDL | 0.019 ± 0.005 | 0.4 ± 0.2 | 0.02 ± 0.02 | 0.06 ± 0.02 | 0.001 ± 0.000 | 4.8 ± 2.6 | 0.004 ± 0.001 | 0.04 ± 0.00 | 0.06 ± 0.03 | 0.3 ± 0.1 |
| Snow over sea ice (n=11) | 290.4 ± 104.5 | 4.3 ± 2.1 | 16.5 ± 5.6 | < MDL | < MDL | 260.8 ± 95.7 | 19.8 ± 9.8 | 5.8 ± 2.2 | 0.02 ± 0.01 | 1.1 ± 0.0 | 0.02 ± 0.01 | 0.07 ± 0.02 | 0.07 ± 0.03 | 0.02 ± 0.00 |
| Sea ice (top 3 cm) (n=4) | 269.0 ± 23.0 | 4.9 ± 1.7 | 17.9 ± 0.8 | < MDL | < MDL | 251.2 ± 20.2 | 18.1 ± 3.0 | 5.0 ± 0.6 | 0.02 ± 0.01 | 1.1 ± 0.0 | 0.02 ± 0.01 | 0.07 ± 0.00 | 0.07 ± 0.01 | 0.02 ± 0.00 |
| Surface seawater (n=3) | 532.1 ± 7.7 | 6.5 ± 1.1 | 28.4 ± 1.1 | < MDL | < MDL | 495.1 ± 1.9 | 33.2 ± 16.6 | 10.3 ± 0.2 | 0.01 ± 0.00 | 1.1 ± 0.0 | 0.01 ± 0.00 | 0.06 ± 0.00 | 0.07 ± 0.03 | 0.02 ± 0.00 |

Another aspect for which NOx is important is that BrONO2 production dominates over HOBr at NO2 > ~100 ppt (depending on HO2 as well) (Wang and Pratt 2017, JGR), which is surely the case for this urban study. This should be added to Figure 1 (the role of BrONO2 in molecular halogen production is also discussed by Wang and Pratt (2017)) and included in the introduction at Lines 44-48.

Response: We will modify our original Figure 1 (shown as Figure 2 in this reply) to include BrONO$_2$ cycling as part of Br reactions. In addition, the dark reduction pathways of Hg(II) will be recognized.

[Figure]

**Figure 2.** General reaction schemes involved in bromine explosion events, ozone depletion events, and mercury depletion events in the Arctic during polar sunrise: gas-phase reactive bromine species (Br and BrO) produced from multi-phase reactions on the surface of the condensed phase will cause the depletion of ozone and gaseous elemental mercury in the boundary layer air (based on (Abbatt et al., 2012; Saiz-Lopez and von Glasow, 2012; Khiri et al., 2020; Simpson et al., 2015, 2007; Wang et al., 2017; Wang and Pratt, 2017). (This figure will be the revised Figure 1 in our revised manuscript.)

In the main text Lines 40-45 (Introduction), the following changes will be made: "While there is a general consensus on the reaction schemes involved in BEEs, ODEs, and MDEs (Fig. 1), major uncertainties exist with respect to the fundamental cryo-photochemical process causing these events and meteorological conditions that may affect their timing and magnitude. It has been generally assumed that the cycling of reactive Br species is sustained by HOBr and BrONO$_2$ via multi-phase reactions on the surface of a condensed phase during polar sunrise (Abbatt et al., 2012; Simpson et al., 2007, 2015; Wang and Pratt, 2017). For instance, the initiation step is thought to be a multi-phase (mp) oxidation of halide

 $\xrightarrow{\text{mp}}$  (R1)

 Yet the role of HOBr and the nature of the condensed phase remain not well characterized."

On Lines 114-115, the authors define "boundary layer air" as "the air mass above the sea ice surface inside the tubes, whereas the air outside of the tubes is considered the "ambient air"", and these terms are then used throughout the manuscript, with the comparison between these air samples being critical to the results. While it is helpful that the authors defined this phrasing in the methods section, it was quite confusing and difficult to remember through the Results & Discussion section, as all air within the boundary layer (not just inside the tubes) would be boundary layer air. I suggest that the authors choose different phrasing that is easier to remember – for example "in-tube air" vs "ambient air".

Response: We agree and the terminology throughout the manuscript will be changed from "boundary layer air" to "in-tube air", which represents the air mass that is constrained inside the acrylic tubes above the sea ice or seawater surface.

Some clarifications of the study conditions are needed. Please directly state in Section 2.1 that the tubes are open to the overlying air. Also, it would be useful to directly state that the sea ice exposure to the atmosphere results in the deposition of atmospheric trace gases and particles to the sea ice. From the comparison in Table 1, however, it is clear that the snow composition (for the ions reported) above the sea ice is dominated by ions from the sea ice brine, based on the comparison to nearby 'land snow'; other ions that would be more impacted by the atmosphere and may be important (e.g. nitrate and nitrite for snow NOx production) should be reported if possible. Also, please clarify in the methods when O3 was measured where (heights and which tubes), as Section 2.2 discusses switching between sampling ports at different heights and locations with 5 min resolution, but then Section 3.2 and Figure 4 seem to show O3 measurements at different heights only occurring on different days. This needs to be very clear if vertical profiles of O3 were not measured on the same day.

Response: In Lines 110-115 (Section 2.1), the following revision will be added: "They were placed about 30 cm away from the edge of the pool and were kept vertical by mechanical arms located on the side of the pool (Fig. 2). Both tubes were open to the overlying atmosphere, which allowed direct air-ice interaction and deposition of atmospheric substances into the sea ice or snowpack surface. One of the tubes was made of UV-blocking acrylic material (cut-off wavelength: 370 nm), and the other of UV-transmitting acrylic material (cut-off wavelength: 270 nm) (see Fig. S1)."

As mentioned in previous reply, nitrate and nitrite concentrations will be included in the revised Table 1.

To resolve the confusion regarding the 5 min resolution and the switching mechanism, we will add more clarifications on the switching mechanisms for ozone measurement in Lines 115-125 (Section 2.2): "To allow ozone measurement of the in-tube  air  between different tubes , a two-way switch Tekran 1100 dual port module was used to automatically switch between the sampling ports on different tubes at an interval of 5 minutes. For example, during 12 to 23 February, continuous sampling was conducted at the same height on UV-transmitting and UV-blocking tubes for more than 40 hours before moving to another height for sampling. And during the time at one height, ozone sampling switched between different tubes at an interval of 5 minutes. Each ozone data reported herein is averaged over a 5-min integration time. The ozone concentrations in the ambient air near the pool were also measured during the experiments."

It would be useful on Lines 84-86 and in Table 1 to report Br-/Cl- ratios to place this in the context of previous studies of Arctic snow Br-/Cl- and the potential to produce Br2 (Peterson et al 2019, Elementa (Figure 5 and associated text); Pratt et al 2013, Nat. Geosc.). Also, it would appear that the Br-/Na+ and Cl-/Na+ labels are reversed in Table1; please fix.

Response: The revised Table 1 is provided in this reply. In the revised manuscript, we will add the following revisions to compare our measured results with the previous studies in Lines 210-215 (Section 3.2): "The mesocosm experiment was conducted using bromide-enriched artificial seawater. As expected, considerable amounts of major ions from seawater are retained in sea ice (Table 1). Major ion concentrations are very low in snow collected from nearby land surfaces, whereas considerably (~ 100
times) higher concentrations are found for the thin layer of snow above sea ice, which is consistent with the brine-wetting process in snow overlying sea ice (Barber and Nghiem, 1999). The measured concentrations of bromide and chloride in snow and surface sea ice samples were much higher than previously reported Arctic snow samples over first-year and multi-year sea ice (Krnavek et al., 2012; Peterson et al., 2019). This large difference can be explained by a dominant contribution from sea ice
brine in our experiment, and a more prevalent atmospheric source of halides in natural Arctic snowpack (Peterson et al., 2019). Bromide is of particular interest, which was found to be preferentially enriched in sea ice and in the overlying snow, as demonstrated by elevated  $Br^-/Cl^-$ molar ratio (0.02) when compared with that in the underlying seawater (0.01). Similar preferential enrichment was not observed for other major ions."

In Lines 290-295 (Section 4.2), the following discussion will be added: "The condensed phase can be either the bare sea ice or the thin layer of snow accumulated on the sea ice surface at the second half of Experiment #1. The observed $Br^-/Cl^-$ ratios in saline snow and in surface sea ice samples are found to be in favor of active Br cycling. Pratt et al (2013) suggested an optimal $Br^-/Cl^-$ mole ratio threshold for $Br_2$ production of 1:200 (0.005), which is below our observed ratios for the potential condensed phase
reactors. However, the highly….."

There are several additional related manuscripts that the authors should consult and incorporate into their manuscript. Nakayama et al. (2015, Tellus, "Ozone depletion in the interstitial air of the seasonal snowpack in northern Japan") is a useful paper for the authors to compare their results to, as they also observed
photochemical O3 loss outside of the Arctic. Helmig et al (2012, JGR, "Ozone dynamics and snow-atmosphere exchanges during ozone depletion events at Barrow, Alaska") is also likely useful, particularly to discuss ozone loss near the surface due to deposition (a topic that should be discussed in this manuscript, but so far is not). Additional important laboratory saline ice studies to cite (especially
on Line 62 when referring to previous frozen halogen release studies; consider whether these are helpful for interpreting results as well) include: Adams et al (2002, Atmos. Chem. Phys., "Uptake and reaction of HOBr on frozen and dry NaCl/NaBr surfaces between 253 and 233K"), Huff and Abbatt (2000, J. Phys. Chem. A, "Gas-phase Br2 production in heterogeneous reactions of Cl2, HOCl, and BrCl
with halideice surfaces"), Wren et al (2013, Atmos. Chem. Phys., "Photochemical chlorine and bromine activation from artificial saline snow"), Halfacre et al (2019, Atmos. Chem. Phys., "pH-dependent production of molecular chlorine, bromine, and iodine from frozen saline surfaces").

Response: We compared our $O_3$ depletion values to Nakayama et al (2015) results, and they are within
the similar magnitude. In the original manuscript, we mentioned there was no obvious $O_3$ deposition on acrylic tubes. This can be supported by the fact that there was no "background" $O_3$ loss observed at 20 cm and 40 cm above the sea ice surface.

Regarding the $O_3$ deposition flux calculation into the snow layer or sea ice surface, there is no $O_3$ measurement available below the surface of the condensed phase. For example, during the entire winter
experiment, no substantial natural snow accumulation (> 5 cm) was observed inside acrylic tubes and the relative distance from sampling port to the ice surface was fixed. These conditions do not support $O_3$ measurement from either snow interstitial air or air trapped within the surface ice sections. Thus, the deposition flux calculation onto the condensed phase (sea ice or snow layer) was not supported in this study.

The following references will be added in the revised manuscript:

Burd, J. A., Peterson, P. K., Nghiem, S. v., Perovich, D. K., and Simpson, W. R.: Snowmelt onset hinders bromine monoxide heterogeneous recycling in the Arctic, 122, https://doi.org/10.1002/2017JD026906, 2017.

Halfacre, J. W., Shepson, P. B., and Pratt, K. A.: PH-dependent production of molecular chlorine, bromine, and iodine from frozen saline surfaces, 19, https://doi.org/10.5194/acp-19-4917-2019, 2019.

Huff, A. K. and Abbatt, J. P. D.: Kinetics and product yields in the heterogeneous reactions of HOBr with ice surfaces containing NaBr and NaCl, 106, 5279–5287, https://doi.org/10.1021/jp014296m, 2002.

Krnavek, L., Simpson, W. R., Carlson, D., Domine, F., Douglas, T. A., and Sturm, M.: The chemical composition of surface snow in the Arctic: Examining marine, terrestrial, and atmospheric influences, 50, https://doi.org/10.1016/j.atmosenv.2011.11.033, 2012.

Nakayama, M., Zhu, C., Hirokawa, J., Irino, T., and Yoshikawa-Inoue, H.: Ozone depletion in the interstitial air of the seasonal snowpack in northern Japan, 67, https://doi.org/10.3402/tellusb.v67.24934, 2015.

Peterson, P. K., Hartwig, M., May, N. W., Schwartz, E., Rigor, I., Ermold, W., Steele, M., Morison, J. H., Nghiem, S. v., and Pratt, K. A.: Snowpack measurements suggest role for multi-year sea ice regions in Arctic atmospheric bromine and chlorine chemistry, 7, https://doi.org/10.1525/elementa.352, 2019.

Wang, S. and Pratt, K. A.: Molecular Halogens Above the Arctic Snowpack: Emissions, Diurnal Variations, and Recycling Mechanisms, 122, https://doi.org/10.1002/2017JD027175, 2017.

Wren, S. N., Donaldson, D. J., and Abbatt, J. P. D.: Photochemical chlorine and bromine activation from artificial saline snow, 13, 9789–9800, https://doi.org/10.5194/acp-13-9789-2013, 2013.

Additional Comments:

Lines 44-47: Note that Pratt et al (2013, Nat. Geosc.) showed that the initiation step of condensed-phase snowpack photochemical production does not require HOBr. Rather HOBr and BrONO2 participate in the bromine explosion cycle that propagates the bromine chemistry. This should be clarified here, as R1 does not represent an 'initiation step' as stated. Dark reaction of O3 with Br- has also been proposed as an initiation step (Artiglia et al. 2017, Nat. Comm.; Simpson et al. 2018, Geophys. Res. Lett.).

Response: The main text will be modified as per our reply above.

Lines 50-52 and 290-291: Note that Pratt et al (2013, Nat. Geosc.) showed directly that Br2 was not produced from sea ice or brine icicles (frost flower proxies) and showed that Br2 production was related to acidity, as supported by lab studies.

Response: We will add more related references to support the statement of acidity requirement in our introduction. The following changes will be made in Lines 50-52: "Yet role of HOBr and the nature of the condensed phase remain not well characterized. Some studies suggested a potential link between the observation of bromine activation with the extent of first-year and multi-year sea ice (Bognar et al., 2020; Simpson et al., 2007) whereas saline snowpack,   and sea salt aerosols have  been proposed to provide such a condensed phase for the reactions by both field observations and laboratory studies (Huff and Abbatt, 2002; Pratt et al., 2013; Wren et al., 2013; Wang et al., 2019).  The cycling of bromine species is favoured under acidic conditions (Pratt et al., 2013; Halfacre et al.,2019) and the surfaces of sea ice and frost flowers, which are highly alkaline (Hare et al., 2013), are unlikely to be effective in  sustaining the reactions."

Lines 52-54: Burd et al. (2017, J. Geophys. Res., "Snowmelt onset hinder bromine monoxide heterogeneous recycling in the Arctic") would be useful to cite here.

Response: We find this paper very relevant and it will be added in the revised reference list. The following change will be made in Line 54: "… rises to above 0 °C (Bognar et al., 2020; Burd et al., 2017; Steffen et al., 2005)."

Lines 81-84: Did the prepared synthetic seawater contain carbonate/bicarbonate? It appears that it did not. Regardless, this should be stated, as it is important for understanding the pH of the sea ice surface, based on the pH dependence of molecular halogen production and the work of Wren and Donaldson (2012, Atmos. Chem. Phys., "How does deposition of gas phase species affect pH at frozen salty interfaces?") that showed that the sea ice surface is buffered against pH change.

Response: We did not intentionally add any carbonate or bicarbonate into the artificial seawater. But the pool was left open for equilibrium with the overlying atmosphere for over a month after the artificial seawater was prepared. Total alkalinity and dissolved inorganic carbon were provided in Table 2. Using the temperature measurement from thermocouples, it allowed us to gain a rough estimate of in-situ bulk pH in surface sea ice sections, which is close to the pH measured on snow and sea ice meltwater for most samples. Unfortunately, direct pH measurements at the air-ice interface were not available in this study.

Lines 192-193 and Table 1: Error should be reported with 1 significant figure throughout the manuscript.

Response: Table 1 will be revised as per our reply above.

Lines 199-201: Please state the absolute magnitude of this [O3] difference here in the text and compare to that observed for the ambient vs boundary layer air (Lines 191-193 and Figure 3).

Response: Lines 199-201 (Section 3.2) will be revised as: "The only exception is the measurement conducted closest to the sea ice surface (10 cm) during the ice-covered period in Experiment #1, when the ozone concentration in the UV-transmitting tube was consistently and considerably lower than that in the UV-blocking tube ($p = 0.00$) (Fig. 4c). The averaged ozone difference between two tubes at 10 cm was $4.6 \pm 2.3$ ppb, whereas the ozone difference between UV-transmitting tube and ambient air was $2.3 \pm 1.9$ ppb for the entire experiment (when applicable)."

Figure 3: Since snow cover is key in this study, can shading be added to this time series to indicate when snow was present?

Response: Unfortunately, we did not have accurate recordings for the snow thickness. During most times, the snow deposition was attached to the inner side of the tube and no substantial snow layer above the sea ice surface was observed.

Line 228: When was the pH of fresh snow measured, and how/where was this snow obtained?

Response: The fresh snow sample was obtained by scooping untouched surface snow on sea ice within 2 hours after deposition, using 50-mL Falcon tubes. Then, the snow samples were melted at 4 °C in dark until the pH measurement. We will add more details in Section 2.3 regarding the snow and ice sampling for pH measurement in the revised manuscript.

---

## Author Comment (AC2)

Dear Referee #2,

Thank you very much for reviewing our manuscript. We appreciate your constructive and valuable comments and suggestions. In the following text, our response to your comments is given in blue, whereas the corresponding revisions in the main text are highlighted in red.

Gao et al. demonstrate that a mesocosm sea-ice facility can be utilized to study bromine explosion events (BEEs) and resultant ozone depletion events (ODEs). The study is to my knowledge an unprecedented demonstration of the utility and potential of mesocosms to study chemistry above sea ice. In particular, the authors are able to compare and contrast bromine and ozone inside two acrylic cylinders one UV-transmitting the other UV-blocking. The authors attribute differences in O3 at 10 cm above the ice surface to a photochemical process being key. Changes in ice and snow conditions and temperature which are largely naturally driven complement this finding by further demonstrating a likely importance of snow and cold temperatures. Where the work can be most improved is that since the work serves primarily as a demonstration of the mesocosm facility and mesocosm experiments generally, certain limitations of this demonstration are inadequately assessed and discussed. In particular the authors note that mercury depletion events (MDE) which typically accompany BEEs can be studied, but this is not yet demonstrated. Further whether the facility replicates organic chemistry and biology which might be relevant to BEEs is not assessed or discussed.

Given that the authors point specifically at using the facility to investigate MDEs in future but have not done so, it would be useful to have some general outline of how such an experiment could be conducted to demonstrate that the mesocosm facility is useful in this regard. This can be very general as much as indicating whether Hg would be a controlled or free variable in such an experiment with some limited detail. Without this limited detail it is difficult to assess if the facility can be used for this purpose as contended.

Response: We thank the referee for recognizing the significance of the novel mesocosm approach we reported here to reproduce Arctic springtime tropospheric chemistry. In this manuscript, the reproduction of ozone depletion events (ODEs) is confirmed by monitoring temporal changes in ozone concentrations above sea ice and relating it to active bromine chemistry (bromine explosion events, BEEs). We expected that the mercury depletion events (MDEs) likely also occurred, but unfortunately, gaseous elemental mercury (GEM) was not measured during the data collection period (2019 to 2020) covered in the original manuscript. In a follow-up experiment conducted during December 2020, both GEM and ozone were measured from the same gas sample line. As expected, MDEs were indeed observed along with ODEs in the same pattern that is consistent with bromine photochemistry (Figure 1). These co-variations between GEM and ozone further demonstrate the capacity of the mesocosm approach in studying photochemical phenomena in the polar troposphere. We will include the data (GEM and ozone) from the follow-up experiment (2020 to 2021) in the revised manuscript.

Our response to the comments on organic chemistry and biology can be found below.

[Figure]

**Figure 1.** Temporal changes of gaseous elemental mercury (GEM) and ozone difference that is calculated between UV-transmitting and UV-blocking tubes during December 2020. (This figure will be added in the revised manuscript)

There are additional variables which might be relevant to BEE which is it unclear if the facility has replicated or can replicate. One is organic carbon content and composition, especially organic carbon present at the surface. Studies have found that the presence of organics can have complex effects on bromide oxidation (Edebeli et al., 2019). In addition, a dark process leading to significant organic bromine from ice and snow in the Antarctic appears to operate under conditions similar to those investigated (Abrahamsson et al., 2018). Can the authors comment on any constraint on the possible effects of organic carbon and whether these are variables the facility can control?

Response: In the mesocosm design, no organic materials were introduced intentionally into the SERF artificial seawater which was prepared freshly from groundwater with NaCl brine and inorganic salts two months before the freezing started. It is possible that traces of organic matter could have entered the outdoor pool (e.g., from dust deposition), but the algal and microbial community growth were minimal due to the cold temperature. As such, we do not think there were any meaningful concentrations of bromocarbons in the experimental seawater/sea ice. Now that we have shown the mesocosm approach is successful, our future research will investigate other processes, including those involving bromocarbons, that may be involved in the occurrence of BEEs/ODEs/MDEs. We will clarify this in the revised manuscript.

A further potential contributing factor are biota. In particular these are likely contributors to ice-nucleation (Irish et al., 2017; Ickes et al., 2020) which can be relevant to the freezing of the simulated sea-water and potentially to the formation of ice particles above the simulated sea or sea-ice surface. Given the identified importance of snow the latter would appear to be especially relevant. Can the authors comment on the microbiome of the microcosm and the capability of the facility to simulate the sea in this regard?

Response: We agree that biota play an important part in the natural cryospheric environment. However, as per our response above, the contribution of biota in the SERF experiments we reported in this manuscript is considered as negligible, since neither biota nor organic matter were added intentionally to the water. This was confirmed in earlier experiments that employed similarly formulated SERF seawater,

where the maximal bulk ice microbial activity and algal chl *a* at the SERF pool were measured to be extremely low ($1.12 \times 10^{-6}$ g CL$^{-1}$ h$^{-1}$ and 0.008 µg L$^{-1}$, respectively) (Gelifus et al., 2016; Rysgaard et al., 2014). We will clarify this in the revised manuscript.

In terms of the role of ice-nucleating particles, it was not explicitly addressed in the original manuscript, but we will add our perspectives in the revised manuscript. We think their potential importance could arise from two processes. First, ice-nucleating particles may affect the roughness of the sea ice surface, which will influence the availability of bromide substrates at the air-ice interface. Second, ice-nucleating particles can assist the formation of frost inside the tube walls, which was observed in early mornings. The frost may serve as a temporary condensed phase medium for the reactions to occur. To which extent these processes can be reproduced at SERF remains uncertain, as both are likely to play a bigger role in a more dynamic ocean environment which is difficult to reproduce in mesocosm experiments. Instead, we think the main advantage of the mesocosm approach is in studying fundamental chemistry and sea ice dynamics.

Regarding this latter point,

I have the following specific comments:

L14: First encountered here but found throughout. I would not use the term "mesocosm scale" or "mesocosm-scale". I am not an expert in mesocosm experiments but I understand them to span several decades of scale (perhaps more) and do not associate them with any inherent scale. In most if not all instances the word "mesocosm" as a noun or adjective would communicate the authors' intent as far as I can tell.

Response: We agree. In the revised manuscript we will change "mesocosm scale" or "mesocosm-scale" to "mesocosm" accordingly.

Figure 1: Realizing this is meant to be a simplified schematic, reactions reducing HgII are not necessarily limited to photolyses, e.g. modeling indicates reduction of HgBrO by CO may be significant (Khiri et al., 2020). Consider adding dashed lines similar to the current photolyses.

Response: We agree. The original Figure 1 will be updated accordingly in the revised manuscript (see below).

[Figure]

**Figure 2.** General reaction schemes involved in bromine explosion events, ozone depletion events, and mercury depletion events in the Arctic during polar sunrise: gas-phase reactive bromine species (Br and BrO) produced from multi-phase reactions on the surface of the condensed phase will cause the depletion of ozone and gaseous elemental mercury in the boundary layer air (based on (Abbatt et al., 2012; Khiri et al., 2020; Saiz-Lopez and von Glasow, 2012; Simpson et al., 2015, 2007; Wang et al., 2017; Wang and Pratt, 2017). (This figure will be the revised Figure 1 in our revised manuscript.)

L45-47: It is very likely that Br- oxidation is the source of volatilized bromine in BEE, however, the central role of HOBr is less certain. In particular, some plausible

alternative oxidants include HOCl (Kumar and Margerum, 1987), HOI (Holmes et al., 2001), and H2O2 (Bray and Livingston, 1928).

Response: We agree that many oxidants can be involved in BEEs and we realized that the HOBr reaction originally present in the manuscript may not be the best representation for the entire reaction series of BEEs. The main text will be rephrased to avoid the confusion and original Figure 1 will be updated.

In the main text Lines 40-45 (Introduction), the following changes will be made: "While there is a general consensus on the reaction schemes involved in BEEs, ODEs, and MDEs (Fig. 1), major uncertainties exist with respect to the fundamental cryo-photochemical process causing these events and meteorological conditions that may affect their timing and magnitude. It has been generally assumed that the cycling of reactive Br species is sustained by HOBr and BrONO2 via multi-phase reactions on the surface of a condensed phase during polar sunrise (Abbatt et al., 2012; Simpson et al., 2007, 2015; Wang and Pratt, 2017). For instance, the initiation step is thought to be a multi-phase (mp) oxidation of halide by HOBr presumably on the surface of a saline condensed phase (Abbatt et al., 2012; Simpson et al., 2007, 2015):

$$X^- + HOBr + H^+ \xrightarrow{mp} H_2O + BrX \tag{R1}$$

where X = Br or Cl. Yet the role of HOBr and the nature of the condensed phase remain not well characterized."

L84-86: The authors note that they have increased the bromide content of the simulated sea water by a factor of roughly eight to enhance the resulting effects and signals. To paraphrase, one function of the mesocosm is to elucidate fundamental processes. It is not known whether BEE are linear with respect to sea water bromide or have some other relation. As such the ability to vary bromide across different mesocosm experiments or even over the course of one experiment would be valuable. The total mass of bromide required, mixing time of the pool, and/or replacement time for the simulated sea water would be helpful in this regard. Can the authors provide this information or point to a relevant reference?

Response: The referee raised a very intriguing question regarding the relationship between BEEs and seawater bromide concentrations, which we do not think has been addressed in the literature. Our mesocosm approach is potentially helpful to carry out such studies in the future.

For the experiments reported in this manuscript, the artificial seawater was prepared by mixing groundwater with NaCl brine and other inorganic salts two months before the start of the winter experiment. During the preparation of artificial water, the dissolution of salts and the mixing of brine were achieved gradually, and no visual observation of solids was confirmed after the preparation of the artificial seawater. In addition, similar salinity measurements were found for waters sampled at different depths in the pool, suggesting a homogeneous distribution of the artificial seawater (no halocline) was reached before the experiment.

Other related SERF experiments are published in the following papers:

Geilfus, N. X., Galley, R. J., Else, B. G. T., Campbell, K., Papakyriakou, T., Crabeck, O., Lemes, M., Delille, B. and Rysgaard, S.: Estimates of ikaite export from sea ice to the underlying seawater in a sea ice-seawater mesocosm, Cryosphere, 10(5), 2173–2189, doi:10.5194/tc-10-2173-2016, 2016.

Hare, A. A., Wang, F., Barber, D., Geilfus, N. X., Galley, R. J. and Rysgaard, S.: PH evolution in sea ice grown at an outdoor experimental facility, Mar. Chem., 154, 46–54, doi:10.1016/j.marchem.2013.04.007, 2013.

Rysgaard, S., Wang, F., Galley, R. J., Grimm, R., Notz, D., Lemes, M., Geilfus, N. X., Chaulk, A., Hare, A. A., Crabeck, O., Else, B. G. T., Campbell, K., Sørensen, L. L., Sievers, J. and Papakyriakou, T.: Temporal dynamics of ikaite in experimental sea ice, Cryosphere, 8(4), 1469–1478, doi:10.5194/tc-8-1469-2014, 2014.

Xu, W., Tenuta, M. and Wang, F.: Bromide and chloride distribution across the snow-sea ice-ocean interface: A comparative study between an Arctic coastal marine site and an experimental sea ice mesocosm, J. Geophys. Res. Ocean., 121(8), 5535–5548, doi:10.1002/2015JC011409, 2016.

L119: "at every minute" could the authors clarify what is meant by this? Is O3 measured as an integration over each minute, or is it a shorter sample taken once each minute.

165    Response: The response time of the ozone instrument (Teledyne T400) is less than 30 s to reach 95 %. The raw ozone file obtained from the instrument was an average over each minute. Since a two-way switch port was used in our gas sample line and the sampling path was switched every 5 minutes between different tubes, each ozone data point discussed throughout the manuscript is an average over five measurements (5 minutes) reported by the instrument from the same sampling port. This will be clarified
170    in the revised manuscript.

   L171: "isothermal" here should be "isotherm"

   Response: Agreed and it will be corrected in the revised manuscript.

   L193: This is the first instance but more follow. Significances for the one-way ANOVA should not be reported as 0.00. They should be reported with the
175    determined significance at higher precision or else as p<0.01.

   Response: In section 2.4, we mentioned that the significant level for the statistical test conducted in this manuscript was set at 0.05, but this is not emphasized in the later text. In the revised manuscript, we will change the expression to either "p > 0.05" or "p < 0.05".

   Fig. 4 c: Certain periods on 2/13 and 2/15 appear to show less difference between
180    the acrylic cylinders, from Fig. 3 these appear to have less shortwave radiation as well. This would seem to be a significant supporting argument for the importance of UV which is not commented on. Could the authors comment?

   Response: We agree that there should be a link between shortwave radiation and the photochemical ozone loss we observed between the two acrylic tubes. Yet this link requires further exploration. At this point,
185    the temporal trend shown in Figure 3 does not support the potential link pointed out by the reviewer. More evidence is required to support that higher shortwave radiation will lead to more photochemical ozone loss. Several other factors may affect this process, including the amount of shortwave radiation (ultraviolet and visible range in particular), the presence of condensed phase reactors (frost, flurries etc), and air temperatures. Thus, more comprehensive mesocosm experiments should be designed to study
190    BEEs. We will mention this in the revised manuscript.

[Figure]

**Figure 3.** Temporal changes of (a) ozone loss (%) between UV-transmitting and UV-blocking tubes; and (b) downward shortwave radiation during 12-15 February 2020. (This figure will be added as supplementary materials in the revised manuscript)

195    Fig. 5 and discussion: The authors comment primarily on pH, however they should also consider the effects of pH on the availability to relevant oxidants and the redox reaction. For instance the pKa of HOBr is 8.59 and HOBr concentrations would change significantly for the pH conditions in this plot.

   Response: The discussion on pH was aimed to provide support for protons that are required in the cycling
200    of reactive bromine species as indicated in the original Figure 1. We agree that the influence of pH on

HOBr dissociation is also an important factor that should be considered. However, one important thing that needs to be clarified is that pH reported in this manuscript is measured on meltwater of bulk snow and sea ice samples, instead of direct pH measurements at the air-ice interface. The dissociation of HOBr will be greatly determined by the pH at the interface, which is not readily available in this study. We will

205   include this in the discussion section of the revised manuscript.

Table 1: The columns for Br-/Na+ and Cl-/Na+ appear to be reversed.

Response: The mistake will be corrected, and the revised Table 1 is provided.

**Table 1.** Major-ion composition of snow, surface ice and surface seawater during Experiment #1

| | Concentration (mmol kg$^{-1}$) | | | | | | | | Molar ratio | | | | | |
|---|---|---|---|---|---|---|---|---|---|---|---|---|---|---|
| | $Cl^-$ | $Br^-$ | $SO_4^{2-}$ | $NO_2^-$ | $NO_3^-$ | $Na^+$ | $Mg^{2+}$ | $Ca^{2+}$ | $Br^-/Cl^-$ | $Cl^-/Na^+$ | $Br^-/Na^+$ | $SO_4^{2-}/Na^+$ | $Mg^{2+}/Na^+$ | $Ca^{2+}/Na^+$ |
| Snow over land (n=8) | 1.5 ± 0.8 | 0.002 ± 0.001 | 0.01 ± 0.00 | < MDL | 0.019 ± 0.005 | 0.4 ± 0.2 | 0.02 ± 0.02 | 0.06 ± 0.02 | 0.001 ± 0.000 | 4.8 ± 2.6 | 0.004 ± 0.001 | 0.04 ± 0.00 | 0.06 ± 0.03 | 0.3 ± 0.1 |
| Snow over sea ice (n=11) | 290.4 ± 104.5 | 4.3 ± 2.1 | 16.5 ± 5.6 | < MDL | < MDL | 260.8 ± 95.7 | 19.8 ± 9.8 | 5.8 ± 2.2 | 0.02 ± 0.01 | 1.1 ± 0.0 | 0.02 ± 0.01 | 0.07 ± 0.02 | 0.07 ± 0.03 | 0.02 ± 0.00 |
| Sea ice (top 3 cm) (n=4) | 269.0 ± 23.0 | 4.9 ± 1.7 | 17.9 ± 0.8 | < MDL | < MDL | 251.2 ± 20.2 | 18.1 ± 3.0 | 5.0 ± 0.6 | 0.02 ± 0.01 | 1.1 ± 0.0 | 0.02 ± 0.01 | 0.07 ± 0.00 | 0.07 ± 0.01 | 0.02 ± 0.00 |
| Surface seawater (n=3) | 532.1 ± 7.7 | 6.5 ± 1.1 | 28.4 ± 1.1 | < MDL | < MDL | 495.1 ± 1.9 | 33.2 ± 16.6 | 10.3 ± 0.2 | 0.01 ± 0.00 | 1.1 ± 0.0 | 0.01 ± 0.00 | 0.06 ± 0.00 | 0.07 ± 0.03 | 0.02 ± 0.00 |

210   L249: The one way ANOVA demonstrates that the UV-transmitting cylinder has lower O3 on average, but not that it is consistently so. See the comment on Fig. 4 on how the consistency is not clear at certain times.

Response: We agree that "consistently" is not appropriate to be used here. In the revised manuscript, we will rephrase the text.

215   L261: I believe this refers to Fig. 4c not Fig. 6c

Response: The mistake will be corrected in the revised manuscript.

---

## Author Response (AR1)

**Reply to Reviewers' Comments (acp-2021-157; Gao et al.)**

We thank two reviewers for their very insightful and constructive comments and suggestions. We have carefully considered all of them and revised the manuscript (ms) accordingly. The revisions are detailed in the track-changes file.

Below we provide a point-by-point response to the reviewers' comments and highlight where changes have been made. The comments from the reviewers are shown in black and **bolded**. Our response is given in the regular font, whereas the corresponding revisions are highlighted in blue. The line numbers in the comments refer to the original manuscript, whereas those in our reply refer to the revised manuscript.

Reviewer #1

**Gao et al present a study of ozone loss near the surface of an outdoor mesocosm sea ice facility in Winnipeg, Canada. The sea ice facility is unique and provides an opportunity to control the formation of the sea ice with complete exposure to the atmosphere, allowing ambient snow accumulation. The comparison of O3 levels during daytime for the UV-transmitting vs UV-blocking tubes is useful. The authors attribute O3 loss at 10 cm above the sea ice surface (observed for the UV-transmitting but not UV-blocking, showing that this is a photochemical process) to reaction with bromine radicals based on their enhancement of seawater Br-, which then migrated into the snow above. My major comment is the lack of discussion of the role of NOx, which is known to be important in O3 and bromine chemistry, and should be particularly important in this urban ambient study (see detailed comments below). This is particularly important because snow photochemical reactions produce NO, which can react with O3, and so disentangling this from reaction with Br is important to consider. Additionally, there is additional published literature, stated below, that is relevant and should be considered in this manuscript.**

**The role of NOx needs to be discussed in this study, as there is currently no mention of NOx in the paper. NO reacts with O3, and it seems possible that this could be contributing in part to the O3 loss observed over snow (see, for example, Peterson and Honrath 2001, Geophys. Res. Lett., "Observations of rapid photochemical destruction of ozone in snowpack interstitial air"), as NOx is released from the snowpack from snow nitrite and nitrate photolysis. Peterson and Honrath (2001) calculated the fraction of the observed ozone loss rate that could be attributed to NOx (they found it was small for their study and then considered bromine reaction), and this seems important to consider here. Were snow nitrate and nitrite measured? If so, this seems important to report. Further, in the urban environment of this study, NOx is likely elevated due to combustion emissions, especially since the authors discuss O3 formation (involving hydrocarbons and NOx) from vehicle exhaust.**

We agree that the role of $NO_x$ should be discussed. Ambient concentrations of $NO_x$ (both NO and $NO_2$) were indeed measured during Experiment #2 from the same gas sample line as the ambient $O_3$ measurement. The concentrations of $NO_x$, $O_3$ as well as $O_3$ loss (%) within the in-tube air (difference between the ambient air and the in-tube air inside the UV-transmitting tube) are provided in Figure S3 in the revised manuscript (see below also). The range for $NO_x$ (NO and $NO_2$) is capped at 100 ppb in Figure S3a, although some extremely high values (up to 300 ppb) were observed due to vehicle activities that were close to the pool. With respect to $NO_x$ influence on ambient $O_3$, the concurrence of NO peaks (Fig S3a) was observed with $O_3$ troughs in the ambient air (Fig. S3b), especially when ambient $O_3$ dropped below 10 ppb. On the other hand, $NO_x$ influence on $O_3$ loss (%) within the in-tube air is less consistent. For example, from 11-14 March, NO peaks (20 ppb) were observed on 11 March and 14 March, whereas NO stayed at relative low levels (< 3 ppb) during 12-13 March. Yet, during the same time (11-14 March), $O_3$ loss (%) within the in-tube air reoccurred daily in a diurnal pattern and reached a similar extent regardless of NO concentrations.

In the original manuscript, this $O_3$ loss (%) within the in-tube air was attributed to the retardant air mixing rates due to the tube effect or enhanced signals from chemical reactions occurred near ice surface. The observed patterns during 11-14 March suggest a more dominant role of photochemical reactions (e.g., Br chemistry) near ice surface and a lesser influence from ambient NO concentrations.

Both nitrate and nitrite were measured in snow samples by ion chromatography as part of the major ion analysis. Nitrite was always below the detection limit (DL) of 4.6 μmol kg$^{-1}$. Nitrate concentrations in most snow samples were also below DL of 1.3 μmol kg$^{-1}$, except for the land snow samples, which were $19 \pm 5$ μmol kg$^{-1}$. Nitrate in land snow is likely sourced from atmospheric deposition, which is originally produced from vehicle exhaust, because nitrate was not intentionally added in our artificial seawater. Due to the very low concentrations of nitrate and nitrite found in the snow above sea ice, it is likely that in-situ $NO_x$ production via snowpack photochemistry is limited within the in-tube air mass.

Furthermore, the photochemical $O_3$ loss we present in the manuscript is obtained by comparing $O_3$ concentrations between the two different acrylic tubes. This comparison helps to cancel out the $O_3$ variations in ambient background (caused either by urban signal or $NO_x$ chemistry) since both tubes were open to the same air mass. Thus, we believe $NO_x$, either from ambient air or snowpack photochemistry, has limited influence on the $O_3$ depletion patterns discussed in this manuscript.

[Figure]

**Figure S3.** Temporal changes of (a) $NO_x$ (NO and $NO_2$) in ambient air; (b) ozone in ambient air, and (c) ozone loss (measured as the difference between in-tube air and ambient air) during Experiment #2.

In addition to the new Figure S3, other revisions are provided below.

Lines 150–153: "In addition to ozone and GEM, nitrogen oxides (NO, $NO_2$ and their sum $NO_x$) concentrations in the ambient air were measured using a Teledyne T200 $NO/NO_2/NO_x$ analyzer near the pool during Experiment #2. The instrument reports data for every minute with a detection limit of 0.4 ppb."

Lines 171-175: "The recovery and detection limit were determined from repetitive measurements on a Dionex seven anion standard and a Dionex six cation-II standard, and were 98 % and 1.5 µmol $kg^{-1}$ for bromide, 96 % and 2.0 µmol $kg^{-1}$ for chloride, 99 % and 3.0 µmol $kg^{-1}$ for sulphate, 97 % and 4.6 µmol $kg^{-1}$ for nitrite, 95 % and 1.3 µmol $kg^{-1}$ for nitrate, 107 % and 0.20 µmol $kg^{-1}$ for sodium, 119 % and 0.8 µmol $kg^{-1}$ for magnesium, and 93 % and 1.0 µmol $kg^{-1}$ for calcium."

Lines 289–315: "In addition, the presence of $NO_x$ would also affect ambient ozone dynamics (Fig. S3) as occasional ozone troughs (near 0 ppb) in the ambient air were observed with NO peaks (Fig. S3a). Similar diurnal patterns of the ozone concentrations are evident in the ambient air and the in-tube air inside the UV-transmitting tube even at 10 cm above the sea ice surface (Fig. 3a and Fig. 7a), suggesting that the ozone concentration in the in-tube air is largely controlled by the urban signal (i.e., the ambient air). However, ozone concentrations in the ambient air are considerably higher than that inside the UV-transmitting tube during sun-lit, daytime (Fig. 7a). This could be indicative of limited mixing of the ambient air inside the tube due to the wall effect, influence on ozone dynamics from $NO_x$ chemistry, and/or loss of ozone inside the tube due to the presence of the experimental sea ice.

To address which of these processes is primarily responsible for the observed ozone difference, we investigate the influence of NOx chemistry and compare the ozone concentrations measured inside the UV-blocking and UV-transmitting tubes. The temporal pattern of the ozone difference between the in-tube air inside the UV-transmitting tube and the ambient air (Fig. S3c) shows the influence of NOx on the ozone loss ($\Delta O_3$) within the in-tube air is limited. For example, from 11 to 14 March, NO peaks (20 ppb) were observed on 11 March and 14 March, whereas NO stayed at relative low levels (< 3 ppb) during 12 to 13 March. Yet, during the same time, $\Delta O_3$ within the in-tube air reoccurred daily in a diurnal pattern and reached a similar extent regardless of NO concentrations. In addition, the in-situ NOx production via snowpack photochemistry of nitrate and nitrite is considered negligible due to the low amount of both ions (below the detection limit) found in surface sea ice and saline snow samples. Thus, $NO_x$ produced either from automobile exhaust or snowpack photochemistry had negligible influence on $\Delta O_3$ within the in-tube air."

Line 352–355: "The other known tropospheric ozone depletion mechanisms include direct photolysis (Wu et al., 2007), $NO_x$ chemistry (Nakayama et al., 2015), and ozone deposition on the tube walls, which are negligible at the study site as demonstrated by a lack of ozone loss during the early fall (Fig. 8c)."

Table 1 has been revised as:

**Table 1.** Ion composition of snow, surface ice and surface seawater during Experiment #1. DL: Detection limit.

| | Concentration (mmol $kg^{-1}$) | | | | | | | | Molar ratio | | | | | |
| --- | --- | --- | --- | --- | --- | --- | --- | --- | --- | --- | --- | --- | --- | --- |
| | $Cl^-$ | $Br^-$ | $SO_4^{2-}$ | $NO_2^-$ | $NO_3^-$ | $Na^+$ | $Mg^{2+}$ | $Ca^{2+}$ | $Br^-/Cl^-$ | $Cl^-/Na^+$ | $Br^-/Na^+$ | $SO_4^{2-}/Na^+$ | $Mg^{2+}/Na^+$ | $Ca^{2+}/Na^+$ |
| Snow over land (n=8) | 1.5 ± 0.8 | 0.002 ± 0.001 | 0.01 ± 0.00 | < DL | 0.019 ± 0.005 | 0.4 ± 0.2 | 0.02 ± 0.02 | 0.06 ± 0.02 | 0.001 ± 0.000 | 4.8 ± 2.6 | 0.004 ± 0.001 | 0.04 ± 0.00 | 0.06 ± 0.03 | 0.3 ± 0.1 |
| Snow over sea ice (n=11) | 290.4 ± 104.5 | 4.3 ± 2.1 | 16.5 ± 5.6 | < DL | < DL | 260.8 ± 95.7 | 19.8 ± 9.8 | 5.8 ± 2.2 | 0.02 ± 0.01 | 1.1 ± 0.0 | 0.02 ± 0.01 | 0.07 ± 0.02 | 0.07 ± 0.03 | 0.02 ± 0.00 |
| Sea ice (top 3 cm) (n=4) | 269.0 ± 23.0 | 4.9 ± 1.7 | 17.9 ± 0.8 | < DL | < DL | 251.2 ± 20.2 | 18.1 ± 3.0 | 5.0 ± 0.6 | 0.02 ± 0.01 | 1.1 ± 0.0 | 0.02 ± 0.01 | 0.07 ± 0.00 | 0.07 ± 0.01 | 0.02 ± 0.00 |
| Surface seawater (n=3) | 532.1 ± 7.7 | 6.5 ± 1.1 | 28.4 ± 1.1 | < DL | < DL | 495.1 ± 1.9 | 33.2 ± 16.6 | 10.3 ± 0.2 | 0.01 ± 0.00 | 1.1 ± 0.0 | 0.01 ± 0.00 | 0.06 ± 0.00 | 0.07 ± 0.03 | 0.02 ± 0.00 |

**Another aspect for which NOx is important is that BrONO2 production dominates over HOBr at NO2 > ~100 ppt (depending on HO2 as well) (Wang and Pratt 2017, JGR), which is surely the case for this urban study. This should be added to Figure 1 (the role of BrONO2 in molecular halogen production is also discussed by Wang and Pratt (2017)) and included in the introduction at Lines 44-48.**

We have modified our original Figure 1 (shown below) to include $BrONO_2$ cycling as part of Br reactions.

[Figure]

**Figure 1.** General reaction schemes involved in bromine explosion events, ozone depletion events and mercury depletion events in the Arctic during polar sunrise. The photochemical activation of gas-phase reactive bromine species (Br and BrO) produced from multi-phase reactions on the surface of the condensed phase causes the depletion of ozone and gaseous elemental mercury in the boundary layer air (based on Abbatt et al., 2012; Khiri et al., 2020; Saiz-Lopez and von Glasow, 2012; Simpson et al., 2007b, 2015; Wang et al., 2017; Wang and Pratt, 2017).

Lines 36–41: the following changes have been made: "While there is a general consensus on the reaction schemes involved in BEEs, ODEs, and MDEs (Fig. 1), major uncertainties exist with respect to the fundamental cryo-photochemical process causing these events and meteorological conditions that may affect their timing and magnitude. It has been generally assumed that the cycling of reactive bromine species is sustained by HOBr and $BrONO_2$ via multi-phase reactions on the surface of a condensed phase during polar sunrise (Abbatt et al., 2012; Simpson et al., 2007, 2015; Wang and Pratt, 2017).

 $\xrightarrow{\text{mp}}$  ⎯⎯⎯⎯⎯⎯⎯⎯⎯⎯⎯⎯⎯ (R1)

 Yet the role of HOBr and the nature of the condensed phase remain not well characterized."

**On Lines 114-115, the authors define "boundary layer air" as "the air mass above the sea ice surface inside the tubes, whereas the air outside of the tubes is considered the "ambient air"", and these terms are then used throughout the manuscript, with the comparison between these air samples being critical to the results. While it is helpful that the authors defined this phrasing in the methods section, it was quite confusing and difficult to remember through the Results & Discussion section, as all air within the boundary layer (not just inside the tubes) would be boundary layer air. I suggest that the**

**authors choose different phrasing that is easier to remember – for example "in-tube air" vs "ambient air".**

We agree and the terminology throughout the manuscript has been changed from "boundary layer air" to "in-tube air", which represents the air mass that is constrained inside the acrylic tubes above the sea ice or seawater surface.

**Some clarifications of the study conditions are needed. Please directly state in Section 2.1 that the tubes are open to the overlying air. Also, it would be useful to directly state that the sea ice exposure to the atmosphere results in the deposition of atmospheric trace gases and particles to the sea ice. From the comparison in Table 1, however, it is clear that the snow composition (for the ions reported) above the sea ice is dominated by ions from the sea ice brine, based on the comparison to nearby 'land snow'; other ions that would be more impacted by the atmosphere and may be important (e.g. nitrate and nitrite for snow NOx production) should be reported if possible. Also, please clarify in the methods when O3 was measured where (heights and which tubes), as Section 2.2 discusses switching between sampling ports at different heights and locations with 5 min resolution, but then Section 3.2 and Figure 4 seem to show O3 measurements at different heights only occurring on different days. This needs to be very clear if vertical profiles of O3 were not measured on the same day.**

In Lines 105-115, the following revision has been made: "They were placed about 30 cm away from the edge of the pool and were kept vertical by mechanical arms located on the side of the pool (Fig. 2). Both tubes were open to the overlying atmosphere, which allowed direct air-ice interaction and deposition of atmospheric substances into the sea ice or snowpack surface. One of the tubes was made of UV-blocking acrylic material (cut-off wavelength: 370 nm), and the other of UV-transmitting acrylic material (cut-off wavelength: 270 nm) (see Fig. S1)."

As mentioned in previous reply, nitrate and nitrite concentrations have been included in the revised Table 1.

To resolve the confusion regarding the 5 min resolution and the switching mechanism, we have added more clarifications on the switching mechanisms for ozone measurement in Lines 125-135: "To allow ozone measurement of the in-tube boundary layer air in between different tubes or at different heights above the surface, a two-way switch (Tekran 1100 dual port module) was used to automatically switch between the sampling ports on different tubes at an interval of 5 minutes. For example, during 12 to 23 February, continuous sampling was conducted at one height on both UV-transmitting and UV-blocking tubes for more than 40 hr before moving to another height for sampling. During data collection at each height, ozone sampling switched between different tubes at an interval of five minutes. Thus, each ozone data reported herein is averaged over a 5-min integration time (five measurements). The ozone concentrations in the ambient air near the pool were also measured during the experiments."

**It would be useful on Lines 84-86 and in Table 1 to report Br-/Cl- ratios to place this in the context of previous studies of Arctic snow Br-/Cl- and the potential to produce Br2 (Peterson et al 2019, Elementa (Figure 5 and associated text); Pratt et al 2013, Nat. Geosc.). Also, it would appear that the Br-/Na+ and Cl-/Na+ labels are reversed in Table1; please fix.**

The revised Table 1 is provided in this reply. In the revised manuscript, we have revised the text to compare our measured results with the previous studies in Lines 252–263: "The mesocosm experiment was conducted using bromide-enriched artificial seawater. As expected, considerable amounts of major ions from seawater are retained in sea ice (Table 1). Major ion concentrations are very low in snow collected from nearby land surfaces, whereas considerably (~ 100 times) higher concentrations are found for the thin

layer of snow above sea ice, which is consistent with the brine-wetting process in snow overlying sea ice (Barber and Nghiem, 1999). The measured concentrations of bromide and chloride in saline snow and surface sea ice samples were much higher than the values previously reported for Arctic snow samples over first-year and multi-year sea ice (Krnavek et al., 2012; Peterson et al., 2019). This large difference can be explained by a dominant contribution from sea ice brine in our mesocosm experiment, and a more prevalent atmospheric source of halides in natural Arctic snowpack (Peterson et al., 2019). Bromide is of particular interest, which was found to be preferentially enriched in sea ice and in the overlying snow, as demonstrated by elevated  $Br^-/Cl^-$ mole ratio (0.02) when compared with that in the underlying seawater (0.01). Similar preferential enrichment was not observed for other major ions."

In Lines 357–361, the following discussion has been added: "The condensed phase can be either the bare sea ice or the thin layer of snow accumulated on the sea ice surface at the second half of Experiment #1 and during Experiment #4. The observed $Br^-/Cl^-$ ratios in saline snow and in surface sea ice samples are found to be in favor of active cycling of bromine species. Pratt et al (2013) suggested an optimal $Br^-/Cl^-$ mole ratio threshold for $Br_2$ production of 0.005, which is below our observed ratios for the potential condensed phase reactors. However, the highly….."

**There are several additional related manuscripts that the authors should consult and incorporate into their manuscript. Nakayama et al. (2015, Tellus, "Ozone depletion in the interstitial air of the seasonal snowpack in northern Japan") is a useful paper for the authors to compare their results to, as they also observed photochemical O3 loss outside of the Arctic. Helmig et al (2012, JGR, "Ozone dynamics and snow-atmosphere exchanges during ozone depletion events at Barrow, Alaska") is also likely useful, particularly to discuss ozone loss near the surface due to deposition (a topic that should be discussed in this manuscript, but so far is not). Additional important laboratory saline ice studies to cite (especially on Line 62 when referring to previous frozen halogen release studies; consider whether these are helpful for interpreting results as well) include: Adams et al (2002, Atmos. Chem. Phys., "Uptake and reaction of HOBr on frozen and dry NaCl/NaBr surfaces between 253 and 233K"), Huff and Abbatt (2000, J. Phys. Chem. A, "Gas-phase Br2 production in heterogeneous reactions of Cl2, HOCl, and BrCl with halideice surfaces"), Wren et al (2013, Atmos. Chem. Phys., "Photochemical chlorine and bromine activation from artificial saline snow"), Halfacre et al (2019, Atmos. Chem. Phys., "pH-dependent production of molecular chlorine, bromine, and iodine from frozen saline surfaces").**

We compared our $O_3$ depletion values to Nakayama et al (2015) results, and they are within the similar magnitude. In the original manuscript, we mentioned there was no obvious $O_3$ deposition on acrylic tubes. This can be supported by the fact that there was no "background" $O_3$ loss observed at 20 cm and 40 cm above the sea ice surface.

Regarding the $O_3$ deposition flux calculation into the snow layer or sea ice surface, there is no $O_3$ measurement available below the surface of the condensed phase. For example, during the entire winter experiment, no substantial natural snow accumulation (> 5 cm) was observed inside acrylic tubes and the relative distance from sampling port to the ice surface was fixed. These conditions do not support $O_3$ measurement from either snow interstitial air or air trapped within the surface ice sections. Thus, the deposition flux calculation onto the condensed phase (sea ice or snow layer) was not supported in this study.

The following references have been added in the revised manuscript:

Burd, J. A., Peterson, P. K., Nghiem, S. v., Perovich, D. K., and Simpson, W. R.: Snowmelt onset hinders bromine monoxide heterogeneous recycling in the Arctic, J. Geophys. Res. Atmos, 122, https://doi.org/10.1002/2017JD026906, 2017.

Halfacre, J. W., Shepson, P. B., and Pratt, K. A.: PH-dependent production of molecular chlorine, bromine, and iodine from frozen saline surfaces, Atmos. Chem. Phys., 19, https://doi.org/10.5194/acp-19-4917-2019, 2019.

Huff, A. K. and Abbatt, J. P. D.: Kinetics and product yields in the heterogeneous reactions of HOBr with ice surfaces containing NaBr and NaCl, 106, 5279–5287, J. Phys. Chem. A, https://doi.org/10.1021/jp014296m, 2002.

Krnavek, L., Simpson, W. R., Carlson, D., Domine, F., Douglas, T. A., and Sturm, M.: The chemical composition of surface snow in the Arctic: Examining marine, terrestrial, and atmospheric influences, Atmos. Environ., 50, https://doi.org/10.1016/j.atmosenv.2011.11.033, 2012.

Nakayama, M., Zhu, C., Hirokawa, J., Irino, T., and Yoshikawa-Inoue, H.: Ozone depletion in the interstitial air of the seasonal snowpack in northern Japan, 67, Tellus B, https://doi.org/10.3402/tellusb.v67.24934, 2015.

Peterson, P. K., Hartwig, M., May, N. W., Schwartz, E., Rigor, I., Ermold, W., Steele, M., Morison, J. H., Nghiem, S. v., and Pratt, K. A.: Snowpack measurements suggest role for multi-year sea ice regions in Arctic atmospheric bromine and chlorine chemistry, 7, Elementa, https://doi.org/10.1525/elementa.352, 2019.

Wang, S. and Pratt, K. A.: Molecular Halogens Above the Arctic Snowpack: Emissions, Diurnal Variations, and Recycling Mechanisms, Atmos. Chem. Phys., 122, https://doi.org/10.1002/2017JD027175, 2017.

Wren, S. N., Donaldson, D. J., and Abbatt, J. P. D.: Photochemical chlorine and bromine activation from artificial saline snow, Atmos. Chem. Phys., 13, 9789–9800, https://doi.org/10.5194/acp-13-9789-2013, 2013.

**Additional Comments:**

**Lines 44-47: Note that Pratt et al (2013, Nat. Geosc.) showed that the initiation step of condensed-phase snowpack photochemical production does not require HOBr. Rather HOBr and BrONO2 participate in the bromine explosion cycle that propagates the bromine chemistry. This should be clarified here, as R1 does not represent an 'initiation step' as stated. Dark reaction of O3 with Br- has also been proposed as an initiation step (Artiglia et al. 2017, Nat. Comm.; Simpson et al. 2018, Geophys. Res. Lett.).**

The text has been revised as per our reply above.

**Lines 50-52 and 290-291: Note that Pratt et al (2013, Nat. Geosc.) showed directly that Br2 was not produced from sea ice or brine icicles (frost flower proxies) and showed that Br2 production was related to acidity, as supported by lab studies.**

We have included more related references to support the statement of acidity requirement in our introduction. The following changes have been made in Lines 40–46: "Yet the role of HOBr and the nature of the condensed phase remain not well characterized. Some studies have suggested a potential link between bromine activation and the extent of first-year and multi-year sea ice (Bognar et al., 2020; Simpson et al., 2007a), whereas field observations and laboratory studies show that saline snowpack and sea salt aerosols are more likely to provide such a condensed phase for the reactions (Huff and Abbatt, 2002; Pratt et al., 2013; Wang et al., 2019; Wren et al., 2013; Xu et al., 2016). The cycling of bromine species is favoured under acidic conditions (Halfacre et al., 2019; Pratt et al., 2013), and the surfaces of sea ice and frost flowers, which are highly alkaline (Hare et al., 2013), are unlikely to be effective in sustaining the reactions."

**Lines 52-54: Burd et al. (2017, J. Geophys. Res., "Snowmelt onset hinder bromine monoxide heterogeneous recycling in the Arctic") would be useful to cite here.**

We find this paper very relevant and it has been added in the revised reference list. The following change has been made in Line 48: "… rises to above 0 °C (Bognar et al., 2020; Burd et al., 2017; Steffen et al., 2005)."

**Lines 81-84: Did the prepared synthetic seawater contain carbonate/bicarbonate? It appears that it did not. Regardless, this should be stated, as it is important for**
**understanding the pH of the sea ice surface, based on the pH dependence of molecular halogen production and the work of Wren and Donaldson (2012, Atmos. Chem. Phys., "How does deposition of gas phase species affect pH at frozen salty interfaces?") that showed that the sea ice surface is buffered against pH change.**

We did not intentionally add any carbonate or bicarbonate into the artificial seawater. But the pool was left open for equilibrium with the overlying atmosphere for over a month after the artificial seawater was prepared. Total alkalinity and dissolved inorganic carbon were provided in Table 2. Using the temperature measurement from thermocouples, it allowed us to gain a rough estimate of in-situ bulk pH in surface sea ice sections, which is close to the pH measured on snow and sea ice meltwater for most samples. Unfortunately, direct pH measurements at the air-ice interface were not available in this study.

The following changes have been made in Lines 89-90: "…. a S = 32.8 seawater. Once prepared, the artificial seawater was left to equilibrate with the atmospheric CO2 for more than one month."

**Lines 192-193 and Table 1: Error should be reported with 1 significant figure throughout the manuscript.**

The manuscript including Table 1 has been revised.

**Lines 199-201: Please state the absolute magnitude of this [O3] difference here in the text and compare to that observed for the ambient vs boundary layer air (Lines 191-193 and Figure 3).**

The following sentence has been added (Lines 229–230): "The averaged ozone difference between the UV-transmitting tube and ambient air was 2.3 ± 1.9 ppb for the entire experiment when measurements were done."

**Figure 3: Since snow cover is key in this study, can shading be added to this time series to indicate when snow was present?**

Unfortunately, we did not have accurate recordings for the snow thickness. During most times, the snow deposition was attached to the inner side of the tube and no substantial snow layer above the sea ice surface was observed.

**Line 228: When was the pH of fresh snow measured, and how/where was this snow obtained?**

The fresh snow sample was obtained by scooping untouched surface snow on sea ice within 2 hours after deposition, using 50-mL Falcon tubes. Then, the snow samples were melted at 4 °C in dark until the pH measurement. We have added more details in Section 2.3 regarding the snow and ice sampling for pH measurement in the revised manuscript (Lines 157–169).

**Reviewer #2:**

**Gao et al. demonstrate that a mesocosm sea-ice facility can be utilized to study bromine explosion events (BEEs) and resultant ozone depletion events (ODEs). The study is to my knowledge an unprecedented demonstration of the utility and potential of mesocosms to study chemistry above sea ice. In particular, the authors are able to compare and contrast bromine and ozone inside two acrylic cylinders one UV-transmitting the other UV-blocking. The authors attribute differences in O3 at 10 cm above the ice surface to a photochemical process being key. Changes in ice and snow conditions and temperature which are largely naturally driven complement this finding by further demonstrating a likely importance of snow and cold temperatures. Where the work can be most improved is that since the work serves primarily as a demonstration of the mesocosm facility and mesocosm experiments generally, certain limitations of this demonstration are inadequately assessed and discussed. In particular the authors note that mercury depletion events (MDE) which typically accompany BEEs can be studied, but this is not yet demonstrated. Further whether the facility replicates organic chemistry and biology which might be relevant to BEEs is not assessed or discussed. Given that the authors point specifically at using the facility to investigate MDEs in future but have not done so, it would be useful to have some general outline of how such an experiment could be conducted to demonstrate that the mesocosm facility is useful in this regard. This can be very general as much as indicating whether Hg would be a controlled or free variable in such an experiment with some limited detail. Without this limited detail it is difficult to assess if the facility can be used for this purpose as contended.**

We thank the referee for recognizing the significance of the novel mesocosm approach we reported here to reproduce Arctic springtime tropospheric chemistry. In this manuscript, the reproduction of ozone depletion events (ODEs) is confirmed by monitoring temporal changes in ozone concentrations above sea ice and relating it to active bromine chemistry (bromine explosion events, BEEs). We expected that the mercury depletion events (MDEs) likely also occurred, but unfortunately, gaseous elemental mercury (GEM) was not measured during the data collection period (2019 to 2020) covered in the original manuscript. In a follow-up experiment conducted during December 2020, both GEM and ozone were measured from the same gas sample line. As expected, MDEs were indeed observed along with ODEs in the same pattern that is consistent with bromine photochemistry (new Figure 4 in the revised manuscript; see below also). These co-variations between GEM and ozone further demonstrate the capacity of the mesocosm approach in studying photochemical phenomena in the polar troposphere. We have included the data (GEM and ozone) from the follow-up experiment (December 2020) in the revised manuscript, and thus changed the title and related sections accordingly.

Our response to the comments on organic chemistry and biology can be found below.

[Figure]

Figure 4. Temporal changes of (a) gaseous elemental mercury (GEM) loss and (b) ozone loss (measured as the difference between the UV-transmitting and UV-blocking tubes), (c) downward shortwave radiation, and (d) surface air temperature during Experiment #4.

**There are additional variables which might be relevant to BEE which is it unclear if the facility has replicated or can replicate. One is organic carbon content and composition, especially organic carbon present at the surface. Studies have found that the presence of organics can have complex effects on bromide oxidation (Edebeli et al., 2019). In addition, a dark process leading to significant organic bromine from ice and snow in the Antarctic appears to operate under conditions similar to those investigated (Abrahamsson et al., 2018). Can the authors comment on any constraint on the possible effects of organic carbon and whether these are variables the facility can control?**

In the mesocosm design, no organic materials were introduced intentionally into the SERF artificial seawater which was prepared freshly from groundwater with NaCl brine and inorganic salts two months before the freezing started. It is possible that traces of organic matter could have entered the outdoor pool (e.g., from dust deposition), but the algal and microbial community growth were minimal due to the cold temperature. As such, we do not think there were any meaningful concentrations of bromocarbons in the experimental seawater/sea ice. Now that we have shown the mesocosm approach is successful, our future research will investigate other processes, including those involving bromocarbons, that may be involved in the occurrence of BEEs/ODEs/MDEs.

We have added the following discussion in the revised manuscript (Line 377–383): "Several improvements can be made for future studies. Even though organic matter and biota were not intentionally introduced into the mesocosm, some of them could have made into the system; to which extent they could have influenced the cryo-photochemical processes (e.g., via the production of bromocarbons) warrants

further exploration. Another aspect is related to ice-nucleating particles which are likely to affect the roughness of the sea ice surface and assist in frost formation on the tube walls during mornings, which could potentially act as a temporary condensed phase for bromine activation just after sunrise. Time-series measurements of pH at the air-ice/snow interface could also be explored to probe the availability of protons and the efficiency of HOBr dissociation on the surface of sea ice or snow layer."

**A further potential contributing factor are biota. In particular these are likely contributors to ice-nucleation (Irish et al., 2017; Ickes et al., 2020) which can be relevant to the freezing of the simulated sea-water and potentially to the formation of ice particles above the simulated sea or sea-ice surface. Given the identified importance of snow the latter would appear to be especially relevant. Can the authors comment on the microbiome of the microcosm and the capability of the facility to simulate the sea in this regard?**

We agree that biota play an important part in the natural cryospheric environment. However, as per our response above, the contribution of biota in the SERF experiments we reported in this manuscript is considered as negligible, since neither biota nor organic matter were added intentionally to the water. This was confirmed in earlier experiments that employed similarly formulated SERF seawater, where the maximal bulk ice microbial activity and algal chl *a* at the SERF pool were measured to be extremely low $(1.12 \times 10^{-6}$ g C L$^{-1}$ h$^{-1}$ and 0.008 µg L$^{-1}$, respectively) (Geilfus et al., 2016; Rysgaard et al., 2014).

In terms of the role of ice-nucleating particles, it was not explicitly addressed in the original manuscript, but we have added our perspectives in the revised manuscript (see our reply above). We think their potential importance could arise from two processes. First, ice-nucleating particles may affect the roughness of the sea ice surface, which will influence the availability of bromide substrates at the air-ice interface. Second, ice-nucleating particles can assist the formation of frost inside the tube walls, which was observed in early mornings. The frost may serve as a temporary condensed phase medium for the reactions to occur. To which extent these processes can be reproduced at SERF remains uncertain, as both are likely to play a bigger role in a more dynamic ocean environment which is difficult to reproduce in mesocosm experiments. Instead, we think the main advantage of the mesocosm approach is in studying fundamental chemistry and sea ice dynamics.

**Regarding this latter point,**
**I have the following specific comments:**
**L14: First encountered here but found throughout. I would not use the term "mesocosm scale" or "mesocosm-scale". I am not an expert in mesocosm experiments but I understand them to span several decades of scale (perhaps more) and do not associate them with any inherent scale. In most if not all instances the word "mesocosm" as a noun or adjective would communicate the authors' intent as far as I can tell.**

We agree. In the revised manuscript we have changed "mesocosm scale" or "mesocosm-scale" to "mesocosm" accordingly.

**Figure 1: Realizing this is meant to be a simplified schematic, reactions reducing HgII are not necessarily limited to photolyses, e.g. modeling indicates reduction of HgBrO by CO may be significant (Khiri et al., 2020). Consider adding dashed lines similar to the current photolyses.**

We agree and Figure 1 has been updated accordingly in the revised manuscript (see our reply to Reviewer #1 above).

**L45-47: It is very likely that Br- oxidation is the source of volatilized bromine in BEE, however, the central role of HOBr is less certain. In particular, some plausible alternative oxidants include HOCl (Kumar and Margerum, 1987), HOI (Holmes et al., 2001), and H2O2 (Bray and Livingston, 1928).**

We agree that many oxidants can be involved in BEEs and we realized that the HOBr reaction originally present in the manuscript may not be the best representation for the entire reaction series of BEEs. We have made the following revisions (Lines 36–41): the following changes have been made: "While there is a general consensus on the reaction schemes involved in BEEs, ODEs, and MDEs (Fig. 1), major uncertainties exist with respect to the fundamental cryo-photochemical process causing these events and meteorological conditions that may affect their timing and magnitude. It has been generally assumed that the cycling of reactive bromine species is sustained by HOBr and BrONO$_2$ via multi-phase reactions on the surface of a condensed phase during polar sunrise (Abbatt et al., 2012; Simpson et al., 2007, 2015; Wang and Pratt, 2017).

$$\text{X}^- + \text{HOBr} + \text{H}^+ \xrightarrow{\text{mp}} \text{H}_2\text{O} + \text{BrX} \qquad \qquad \text{(R1)}$$

 Yet the role of HOBr and the nature of the condensed phase remain not well characterized."

**L84-86: The authors note that they have increased the bromide content of the simulated sea water by a factor of roughly eight to enhance the resulting effects and signals. To paraphrase, one function of the mesocosm is to elucidate fundamental processes. It is not known whether BEE are linear with respect to sea water bromide or have some other relation. As such the ability to vary bromide across different mesocosm experiments or even over the course of one experiment would be valuable. The total mass of bromide required, mixing time of the pool, and/or replacement time for the simulated sea water would be helpful in this regard. Can the authors provide this information or point to a relevant reference?**

The referee raised a very intriguing question regarding the relationship between BEEs and seawater bromide concentrations, which we do not think has been addressed in the literature. Our mesocosm approach is potentially helpful to carry out such studies in the future.

For the experiments reported in this manuscript, the artificial seawater was prepared by mixing groundwater with NaCl brine and other inorganic salts two months before the start of the winter experiment. During the preparation of artificial water, the dissolution of salts and the mixing of brine were achieved gradually, and no visual observation of solids was confirmed after the preparation of the artificial seawater. In addition, similar salinity measurements were found for waters sampled at different depths in the pool, suggesting a homogeneous distribution of the artificial seawater (no halocline) was reached before the experiment.

Other related SERF experiments are published in the following papers:

Geilfus, N. X., Galley, R. J., Else, B. G. T., Campbell, K., Papakyriakou, T., Crabeck, O., Lemes, M., Delille, B. and Rysgaard, S.: Estimates of ikaite export from sea ice to the underlying seawater in a sea ice-seawater mesocosm, Cryosphere, 10(5), 2173–2189, doi:10.5194/tc-10-2173-2016, 2016.

Hare, A. A., Wang, F., Barber, D., Geilfus, N. X., Galley, R. J. and Rysgaard, S.: PH evolution in sea ice grown at an outdoor experimental facility, Mar. Chem., 154, 46–54, doi:10.1016/j.marchem.2013.04.007, 2013.

Rysgaard, S., Wang, F., Galley, R. J., Grimm, R., Notz, D., Lemes, M., Geilfus, N. X., Chaulk, A., Hare, A. A., Crabeck, O., Else, B. G. T., Campbell, K., Sørensen, L. L., Sievers, J. and Papakyriakou, T.:

Temporal dynamics of ikaite in experimental sea ice, Cryosphere, 8(4), 1469–1478, doi:10.5194/tc-8-1469-2014, 2014.

Xu, W., Tenuta, M. and Wang, F.: Bromide and chloride distribution across the snow-sea ice-ocean interface: A comparative study between an Arctic coastal marine site and an experimental sea ice mesocosm, J. Geophys. Res. Ocean., 121(8), 5535–5548, doi:10.1002/2015JC011409, 2016.

**L119: "at every minute" could the authors clarify what is meant by this? Is O3 measured as an integration over each minute, or is it a shorter sample taken once each minute.**

The response time of the ozone instrument (Teledyne T400) is less than 30 s to reach 95 %. The raw ozone file obtained from the instrument was an average over each minute. Since a two-way switch port was used in our gas sample line and the sampling path was switched every 5 minutes between different tubes, each ozone data point discussed throughout the manuscript is an average over five measurements (5 minutes) reported by the instrument from the same sampling port.

This has been clarified in the revised manuscript (Line 128–134): "To allow ozone measurement of the in-tube  air mass  between different tubes , a two-way switch Tekran 1100 dual port module was used to automatically switch between the sampling ports on different tubes at an interval of 5 minutes. For example, during 12 to 23 February, continuous sampling was conducted at one height on both UV-transmitting and UV-blocking tubes for more than 40 hr before moving to another height for sampling. During data collection at each height, ozone sampling switched between different tubes at an interval of five minutes. Each ozone data reported herein is averaged over a 5-min integration time. The ozone concentrations in the ambient air near the pool were also measured during the experiments."

**L171: "isothermal" here should be "isotherm"**

Agreed and it has been corrected in the revised manuscript.

**L193: This is the first instance but more follow. Significances for the one-way ANOVA should not be reported as 0.00. They should be reported with the determined significance at higher precision or else as p<0.01.**

In section 2.4, we mentioned that the significant level for the statistical test conducted in this manuscript was set at 0.05, but this is not emphasized in the later text. In the revised manuscript, we have changed the expression to either "p > 0.05" or "p < 0.05".

**Fig. 4 c: Certain periods on 2/13 and 2/15 appear to show less difference between the acrylic cylinders, from Fig. 3 these appear to have less shortwave radiation as well. This would seem to be a significant supporting argument for the importance of UV which is not commented on. Could the authors comment?**

We agree that there should be a link between shortwave radiation and the photochemical ozone loss we observed between the two acrylic tubes. Yet this link requires further exploration. At this point, the temporal trend shown in Figure 3 does not support the potential link pointed out by the reviewer, as shown in a new Figure S4 in the revised manuscript (see below also). More evidence is required to support that higher shortwave radiation will lead to more photochemical ozone loss. Several other factors may affect this process, including the amount of shortwave radiation (ultraviolet and visible range in particular), the presence of condensed phase reactors (frost, flurries etc), and air temperatures. Thus, more comprehensive mesocosm experiments should be designed to study BEEs.

We have included this discussion in the revised manuscript (Line 343–348): "It should also be noted that the relationship between the intensity of downward shortwave radiation and the extent of the cryo-photochemical ozone loss is yet to be validated (Fig. 4 and Fig. S4). The lack of correlation may be explained by the fact that the downward shortwave radiation reported in this study covers wavelength from 300 nm to 2800 nm, whereas UV radiation (< 400 nm) has been generally considered the most effective range causing photochemical ozone loss. Other factors, such as the presence of various condensed phase, the availability of Br– substrates and air temperatures would also affect the extent of the cryo-photochemical ozone loss."

[Figure]

Figure S4. Temporal changes of (a) ozone loss (%) between the UV-transmitting and UV-blocking tubes; and (b) downward shortwave radiation during Experiment #4.

**Fig. 5 and discussion: The authors comment primarily on pH, however they should also consider the effects of pH on the availability to relevant oxidants and the redox reaction. For instance the pKa of HOBr is 8.59 and HOBr concentrations would change significantly for the pH conditions in this plot.**

The discussion on pH was aimed to provide support for protons that are required in the cycling of reactive bromine species as indicated in the original Figure 1. We agree that the influence of pH on HOBr dissociation is also an important factor that should be considered. However, one important thing that needs to be clarified is that pH reported in this manuscript is measured on meltwater of bulk snow and sea ice samples, instead of direct pH measurements at the air-ice interface. The dissociation of HOBr will be greatly determined by the pH at the interface, which is not readily available in this study. We have included this in the revised manuscript (Lines 382–383): "Time-series measurements of pH at the air-ice/snow interface could also be explored to probe the availability of protons and the efficiency of HOBr dissociation on the surface of sea ice or snow layer."

**Table 1: The columns for Br-/Na+ and Cl-/Na+ appear to be reversed.**

The mistake has been corrected; see the revised Table 1 in our response to Reviewer #1 above.

**L249: The one way ANOVA demonstrates that the UV-transmitting cylinder has lower O3 on average, but not that it is consistently so. See the comment on Fig. 4 on how the consistency is not clear at certain times.**

We agree that "consistently" is not appropriate to be used here. In the revised manuscript, we have rephrased the text in Lines 307-310: "The overall temporal patterns between the two tubes were similar during Experiment #1 (Fig. 5). Generally, the ozone concentration in the boundary layer air immediately (10 cm) above the sea ice in the UV-transmitting tube were considerably lower than those in the UV-blocking tube (Fig. 5c)."

**L261: I believe this refers to Fig. 4c not Fig. 6c**

The mistake has been corrected in the revised manuscript.

---

## Author Response (AR2)

**Reply to Reviewers' Comments**

We thank Editor Dr. Jayan Kuttippurath and two reviewers for their comments on an earlier version of the manuscript. We have carefully considered all the comments and revised the manuscript accordingly.

Below we provide a detailed, point-by-point list of our reply to all the comments. The comments from the reviewers are shown in **bold** text, and our responses in normal text. We also show in blue the revised text as it appears in the revised manuscript. The page and line numbers in the reviewers' comments refer to the previous version, whereas those in this reply refer to the latest revised manuscript.

**Reviewer #1:**

**[No further comments. Recommended the manuscript be accepted as is.]**

**Reviewer #2:**

**Gao et al present a revision of their manuscript describing observations of ozone loss above snow in an outdoor mesocosm experiment. The change of phrasing to "in-tube air" and "boundary layer air", in particular, is very helpful in clarifying the discussion and results. The addition of the GEM measurements in Dec 2020 and associated simultaneously observed GEM and $O_3$ depletion is highly compelling for the role of Br chemistry and a great addition to the manuscript. As discussed below, the manuscript would be further significantly strengthened by quantitative calculations of the contribution of NO titration by $O_3$ (when $NO_x$ data were available) to indirectly, quantitatively calculate the expected contribution by Br (see below for details). Line numbers below refer to the track changes version of the manuscript.**

**In response to the reviewer #1 point about likely $NO_x$ influence, the authors added NO and $NO_2$ ambient observational data, which is excellent, in Figure S3, which also shows ozone in the ambient air (ppb), and ozone loss (%) within in-tube air. $NO_x$ often increased to 30-40 ppb, and sometimes higher, and during these periods $O_3$ typically declined by 10-30 ppb, as shown in Figure S3. The largest ozone losses (%) within the in-tube air appeared to occur simultaneously with the ambient ozone reductions and $NO_x$ peaks (e.g. Mar 4, 6, 9, 10, 12). The authors state, however, in their response that "we believe $NO_x$, either from ambient air or snowpack photochemistry, has limited influence on the $O_3$ depletion patterns discussion in this manuscript." (Similar wording is presented in the new text on Lines 310-317 of the manuscript.) However, no calculations are presented to quantify the $O_3$ loss due to NO titration in the in-tube air to support this statement, and the data in the Figure S3 do not appear to support the authors' assertion of a lack of temporal correlation between $NO_x$ and $O_3$ loss. Such calculations are necessary to quantify the loss of ozone by reaction with NO (which is photochemically produced) given the common correlation between $O_3$ loss and $NO_x$ in the ambient air, as well as the very high $[NO_x]$ levels observed in this work. If a large fraction of the $O_3$ loss cannot be accounted for by NO titration, this would quantitatively**

**support the authors' assertion. Examination of the O₃ loss rates for each depletion period would also be helpful for this evaluation and useful for quantifying the role of Br. It would also be helpful for the absolute [O₃] within the in-tube air to be included in Figure S3.**

We agree and have made several revisions to clarify the role of $NO_x$ chemistry in the $O_3$ dynamics. First, the original Fig. S3 has been updated (see Fig. 1 in this reply). The $NO_x$ graph has been changed to have the same resolution (10-minute) as the $O_3$ loss measurements, instead of the one-minute resolution in the previous version. Absolute $O_3$ loss (ppbv) has also been provided, which shows a very similar trend as the normalized $O_3$ loss (%). The use of absolute $O_3$ loss removes some overestimated $O_3$ loss (%) points when the ambient $O_3$ is mostly depleted (< 5 ppbv) (e.g., on March 4, 6 and 11).

For $O_3$ in the ambient air, ambient $O_3$ troughs were often observed with NO peaks, most of which were during night (20:00 to 7:00; local time) when $O_3$ in the ambient air was largely depleted (< 10 ppbv). This observation can be explained by the NO + $O_3$ reaction and the ambient air dynamics during night. Occasionally, a sharp increase of $NO_x$ in the ambient air and a small scale of $O_3$ depletion in both the ambient air and in-tube air were observed during daytime, which are most likely caused by the occasional use of vehicles within the SERF facility (e.g., 12:00 on March 6 and 9, see Fig. 2).

In this manuscript, we studied the ozone loss ($\Delta O_3$) by comparing between different locations (the ambient air vs. in-tube air; the UV-transmitting tube vs. UV-blocking tube). For $\Delta O_3$ within the in-tube air (the ambient air vs. in-tube air), the ambient air can be considered as a control group while the in-tube air is the experimental group that examines the influence from sea ice and the acrylic tube. Since the tube was open to the ambient environment, we consider similar air mass for both the ambient air and in-tube air, including $NO_x$ from urban signal that was already present in the background ambient air. This assumption is also supported by overall similar $O_3$ trends observed between the ambient air and in-tube air (see Fig. 3a in the revised manuscript and Fig. 2 in this reply). In this case, NO titration (mainly produced from urban signals) would equally affect $O_3$ dynamics in the ambient air and in-tube air, which is an offset process when studying $\Delta O_3$ via the comparison between the ambient air and in-tube air. Thus, such $\Delta O_3$ should not be affected by variations of $NO_x$ in the background ambient air. Since the $\Delta O_3$ within the in-tube air examines the influence from sea ice, $NO_x$ production via snowpack photochemistry may be important, yet this process is considered negligible due to the low concentrations of nitrite and nitrate found in surface ice and saline snow samples as discussed in the manuscript. Then, $NO_x$ that may influence $O_3$ dynamics is only expected from urban signals (the background ambient air) and should have little influence on the $\Delta O_3$ within the in-tube air. Furthermore, when we compare $O_3$ between the UV-transmitting and UV-blocking tubes, we consider both in-tube air masses are similar and background $NO_x$ would affect both to a similar extent whereas the only variable being examined is the UV radiation. We thus believe our general assumption that variations of $NO_x$ in the background ambient air do not affect $\Delta O_3$ obtained from comparisons is sound.

The assumption that the air mass between the ambient air and in-tube air was similar is challenged when there was a sudden and rapid disturbance (e.g., occasional use of vehicles within the facility). In this case, the vehicle exhaust signal was readily captured in the ambient air, whereas the in-tube air was less affected due to a lack of rapid air mixing. During daytime, $O_3$ can be produced from oxidation of hydrocarbons in the vehicle exhaust or $NO_2$ photochemistry. This increased $O_3$ signal (within 20 minutes from the vehicle use) could be immediately observed as a

lesser extent of O₃ depletion in the ambient air at the beginning of the disturbance, whereas such O₃ increase was not necessarily captured in the in-tube air due to a lack of rapid air mixing. This condition would overestimate $\Delta O_3$ and result in those abnormal high values out of the general $\Delta O_3$ trend observed around 12:00 on March 6 and 9 (shaded areas in Fig. 2). NO produced from on-site use of vehicles could subsequently cause small-scale O₃ depletion in both the ambient air and in-tube air. However, during each day, the $\Delta O_3$ was observed before the sharp increase of $NO_x$ and continued until sunset, indicating that daytime $NO_x$ production due to on-site use of vehicles was not the main driver for $\Delta O_3$ within the in-tube air.

[Figure]

**Figure 1.** Temporal changes of (a) $NO_x$ (NO + $NO_2$) in the ambient air; (b) ozone in the ambient air; (c) ozone loss (%) and (d) ozone loss (ppbv) (measured as the difference between the ambient air and in-tube air) during Experiment #2. (This figure has been added as Figure S3 in the Supplementary Information)

[Figure]

**Figure 2.** Temporal changes of $O_3$, NO, $NO_2$, and $O_3$ loss within the in-tube air (measured as the difference between the ambient air and in-tube air) during each major ambient $O_3$ depletion event of Experiment #2. (This figure has been added as Figure S4 in the Supplementary Information)

The reviewer pointed out that the largest $\Delta O_3$ (%) within the in-tube air sometimes coincides with NO peaks and ambient $O_3$ depletions. After examining each ambient $O_3$ depletion period (< 20 ppbv, March 4, 6, 9, 10, 11 and 15) thoroughly with absolute $\Delta O_3$ (ppbv), we think such temporal coincidence is not prevalent (Fig. 2). Major ambient $O_3$ depletions and NO peaks were mostly observed during night except those small scale $O_3$ depletions during daytime (see discussion above). On the other hand, $\Delta O_3$ showed up during daytime with a diurnal pattern and the duration is generally longer than the occasional daytime NO peaks. Meanwhile, the maximal

extent of $\Delta O_3$ (ppbv) from March 9 to 16 was similar regardless of the variations of $NO_x$ in the ambient air. Moreover, the correlation between NO concentrations and $\Delta O_3$ (ppbv) or $\Delta O_3$ (%) is examined and the correlation coefficient (r) is –0.18 and 0.13, respectively, suggesting no strong correlation.

Unfortunately, we currently do not have all the parameters to quantify the NO influence using a model simulation. However, the contribution from $NO_x$ chemistry on depleting ozone can be qualitatively estimated using the $NO/NO_2$ ratio. When the total amount of $NO_x$ (NO + $NO_2$) remains relative stable and without high concentrations of volatile organic compounds, the $NO/NO_2$ ratio would decrease if the $NO + O_3$ reaction proceeds to any substantial extent; otherwise, it would increase (Finlayson-Pitts and Pitts, 2000). In Fig. 3, typical daytime patterns are provided, which shows a distinct behavior of $NO_x$ chemistry: the $NO/NO_2$ ratio increased on March 5 and 14 whereas it decreased on March 11 while $NO_x$ stayed at a relative stable level during the observed $\Delta O_3$ span. Based on our experimental design, $\Delta O_3$ should not be attributed to $NO_x$ in the ambient air, but if we assume $NO_x$ chemistry-driven $O_3$ depleting process does contribute to $\Delta O_3$ within the in-tube air, a more important contribution from $NO_x$ chemistry would be expected on March 11 and other processes (e.g., Br chemistry) would have contributed to the $\Delta O_3$ on March 5 and 14. Such examinations are carried out on each daytime $\Delta O_3$ period when $NO_x$ measurements were available (March 3 to 16), and the potential $NO_x$ contribution to $\Delta O_3$ (decreasing $NO/NO_2$ ratio during the observed $\Delta O_3$ span) was only observed on two days (March 3 and 11).

[Figure]

**Figure 3.** Temporal changes of ambient $O_3$, NO, $NO_x$, and $O_3$ loss within the in-tube air (measured as the difference between the ambient air and in-tube air) and $NO/NO_2$ ratio during the daytime $O_3$ loss span on March 5 (a, b), March 11 (c, d) and March 14 (e, f). (This has been added as Figure S5 in the Supplementary Information)

There is a possiblity that the $\Delta O_3$ between the UV-transmitting and UV-blocking tubes can be affected by photochemically active $NO_x$ chemistry, especially intiated by UV radiation. This influence, if possible, is expected to occur throughout the arcylic tube and regardless of the sea ice presence. However, the obervation that no such $O_3$ loss was observed at 20 cm and 40 cm above the sea ice surface or above open water surface suggests this process has minimal contribution to the cryo-photochemical ozone loss we report in the manuscript.

In conclusion, the ambient $O_3$ dynamics is associated with NO concentrations, especially during ambient $O_3$ depletions over night. However, $\Delta O_3$ obtained from comparisons should not be attributed to variations of $NO_x$ in the background ambient air by the experimental design and the in-situ $NO_x$ production via snowpack photochemistry is considered negligible. Thus, $NO_x$ produced either from the urban signal or snowpack photochemistry has limited influence on the $\Delta O_3$ within the in-tube air. Still, if we assume $NO_x$ chemistry does contribute to $\Delta O_3$ within the in-tube air, the potential contribution is only indicated by a decreasing $NO/NO_2$ ratio on two days during the two-week Experiment #2. Since the general diurnal pattern of $\Delta O_3$ cannot be explained by $NO_x$ chemistry, we believe Br chemistry is most likely the main driver for $\Delta O_3$ reported in this manuscript.

The related discussion has been added in the Supplementary Information.

In lines 309-315, the main text has been revised as: "*Yet, during the same time, $\Delta O_3$ within the in-tube air reoccurred daily in a diurnal pattern and reached a similar extent regardless of NO concentrations. Additional discussion and figures on the potential influence of $NO_x$ chemistry on $\Delta O_3$ within the in-tube air can be found in the Supplementary Information. On the other hand, the in-situ $NO_x$ production via snowpack photochemistry of nitrate and nitrite is considered negligible due to the low amount of both ions (below the detection limit) found in surface sea ice and saline snow samples. Thus, $NO_x$ produced either from urban transportation (in the background ambient air) or snowpack photochemistry had negligible influence on $\Delta O_3$ within the in-tube air (i.e., ozone difference between the ambient air and in-tube air inside the UV-transmitting tube).*"

In lines 336-340, the main text has been revised as: "*The observation that no such ozone loss occurred in the in-tube air when measured farther away (20 cm and 40 cm) from the sea ice surface (Fig. 5a, b) suggests that the ozone loss between two tubes is not associated with $NO_x$ photochemistry or ozone photolysis, which should occur universally throughout the tube. Instead, it is most likely triggered by cryo-photochemical processes that involve the sea ice environment.*"

In lines 350-353, the main text has been revised as: "*Moreover, comparing ozone concentrations between tubes cancels out the variations that exist in the ambient air (e.g., background $NO_x$ produced from urban transportation) since both tubes were open to the same air mass and the ozone dynamics in the ambient air should be equally applied to both tubes.*"

**Additional Comments:**

**Lines 58-59: Kalnajs and Avallone (2006, Geophys. Res. Lett) is an important original reference for the suggestion that the pH of frost flowers is too high to support reactive bromine chemistry.**

This paper has been added as a reference in the revised manuscript.

**Lines 178-182 and Table 1 caption: Please define what the detection limit is here, since several data points in Table 1 are below the detection limit. Is this the 3*sigma of nanopure water, for example?**

The detection limit was determined from 8 repetitive measurements on the least concentrated point of the calibration curve (28 µmol kg$^{-1}$ chloride for anions and 4.3 µmol kg$^{-1}$ sodium for cations, respectively). The detection limit was calculated as 2.998 × standard deviation of the repetitive measurements.

In lines 174-176, the main text has been revised as: *"The recovery and detection limit were determined from repetitive measurements on the least concentrated point of the calibration curve, prepared from a Dionex seven anion standard and a Dionex six cation-II standard respectively"*.

Table 1 title has been revised accordingly: *"**Table 1.** Ion composition of snow, surface ice and surface seawater during Experiment #1. DL: Detection limit calculated as 2.998 times the standard deviation determined from eight repetitive measurements on the least concentrated point of the calibration curve"*.

**Figure 1: Note that BrONO2 is also thought to undergo hydrolysis in the condensed phase to produce HOBr, which then reacts to produce BrX (Aguzzi and Rossi 2002, J Phys Chem A). Also, in the caption, did the authors mean to cite Wang et al 2019 (in their reference list) instead of Wang et al 2017 here, since Wang et al 2019 (PNAS) was the first to quantitatively observe O3 and Hg(0) loss via reaction with Br? Likewise, Wang et al 2019 (PNAS) would be useful to cite on Lines 49-51 because that work included measurements of Br, BrO, Br2, and HOBr.**

Figure 1 has been modified (see next page). The BrONO$_2$ hydrolysis reaction that produces HOBr on the condensed phase has been added. The reduction pathway of Hg$^{II}$ to Hg$^0$ in the gas phase has been updated with a recognition of the potential role of Hg$^I$ during the reduction. In the caption, Wang et al. (2017) (Sea Ice Book chapter) is cited as a reference for Hg chemical cycles in the polar environment. The Aguzzi and Rossi paper, Wang et al. 2019 (PNAS) paper and Saiz-Lopez et al. (2018, 2019) papers on Hg$^{II}$ and Hg$^I$ reduction have also been added as additional references.

**Figure 8: There appears to be a typo in the caption, as part b is referred to as both ice-covered and open water. Please fix. I am assuming that the authors mean to write "open water (c, d) experiments".**

This mistake has been corrected in the revised manuscript.

**I am confused by the authors statement on Page 8 of their response that "no substantial snow layer above the sea ice surface was observed" (in response to the request to add shading for when snow cover was present) because the methods section and elsewhere discusses snow composition, and Lines 222-223, for example, state "…up to 4 cm of natural snow accumulation inside the tubes were observed above sea ice…".**

Regarding the shading for snow cover presence, we did not record the temporal change of the snow depth and structure; instead, we only measured the depth of the snow right after the precipitation. Thus, we cannot show the exact duration for the presence of the snow cover but only the thickness of the snow cover that was measured at one time.

[Figure]

**Figure 4.** General reaction schemes involved in bromine explosion events, ozone depletion events and mercury depletion events in the Arctic during polar sunrise. The photochemical activation of gas-phase reactive bromine species (Br and BrO) produced from multi-phase reactions on the surface of the condensed phase causes the depletion of ozone and gaseous elemental mercury in the boundary layer air (based on Abbatt et al., 2012; Aguzzi and Rossi, 2002; Khiri et al., 2020; Saiz-Lopez et al., 2018, 2019; Saiz-Lopez and von Glasow, 2012; Simpson et al., 2007b, 2015; F.Wang et al., 2017; S.Wang et al., 2019; Wang and Pratt, 2017). (This figure is the new revised Figure 1 in the revised manuscript)

**Lines 371-372: To clarify, consider rephrasing sentence as "Pratt et al. (2013) and Peterson et al. (2019) suggested that snow Br2 production is enhanced above a Br-/Cl- mole ratio of 0.005."**

The text has been revised as suggested to avoid confusion.

In lines 367-368: *"Pratt et al. (2013) and Peterson et al. (2019) suggested $Br_2$ production from snow is enhanced above a  $Br^-/Cl^-$ mole ratio  of 0.005"*.

**Significant Figures in Reported Errors: The authors state in their response that they fixed the reporting of errors to be 1 significant figure throughout the manuscript, but at a quick glance, that does not appear to be the case throughout (e.g., Lines 229-231).**

For $NO_x$ and $O_3$ measurements, the instrument detection limit is 0.4 ppbv, and we report numbers above that and keep accuracy to 1 ppbv. For ion analysis, chloride and sodium results are

reported to the first digit for most samples except for "snow over land"; for the rest ions, the results are reported to be first decimal point except for "snow over land".

**Additional Revisions**

In addition to the reviewer's comments, we have made the following revisions to the manuscript:

1) The unit for $O_3$ and $NO_x$ data has been changed from ppb to ppbv.

2) Clarifications on the resolution of $NO_x$ and ozone loss measurement have been added in lines 138-139: "*The quantify and normalize the ozone difference inside the UV-transmitting tube relative to other locations, the ozone loss (%) is reported for every 10 minutes and calculated by Eq (1):*"

3) In lines 154-156, the text has been revised as: "*The instrument reports data for every minute with a detection limit of 0.4 ppbv. The $NO_x$ data reported in this study is averaged for every 10 minutes, which is the same resolution as the ozone loss ($\Delta O_3$) measurement.*"